# ADOPT: Modified Adam Can Converge with Any $\beta_2$ with the Optimal Rate

**Shohei Taniguchi**
The University of Tokyo
taniguchi@weblab.t.u-tokyo.ac.jp

**Keno Harada**
The University of Tokyo
keno.harada@weblab.t.u-tokyo.ac.jp

**Gouki Minegishi**
The University of Tokyo
minegishi@weblab.t.u-tokyo.ac.jp

**Yuta Oshima**
The University of Tokyo
yuta.oshima@weblab.t.u-tokyo.ac.jp

**Seong Cheol Jeong**
The University of Tokyo
jeong@weblab.t.u-tokyo.ac.jp

**Go Nagahara**
The University of Tokyo
nagaharago@weblab.t.u-tokyo.ac.jp

**Tomoshi Iiyama**
The University of Tokyo
iiyama@weblab.t.u-tokyo.ac.jp

**Masahiro Suzuki**
The University of Tokyo
masa@weblab.t.u-tokyo.ac.jp

**Yusuke Iwasawa**
The University of Tokyo
iwasawa@weblab.t.u-tokyo.ac.jp

**Yutaka Matsuo**
The University of Tokyo
matsuo@weblab.t.u-tokyo.ac.jp

## Abstract

Adam is one of the most popular optimization algorithms in deep learning. However, it is known that Adam does not converge in theory unless choosing a hyperparameter, i.e., $\beta_2$, in a problem-dependent manner. There have been many attempts to fix the non-convergence (e.g., AMSGrad), but they require an impractical assumption that the gradient noise is uniformly bounded. In this paper, we propose a new adaptive gradient method named ADOPT, which achieves the optimal convergence rate of $\mathcal{O}(1/\sqrt{T})$ with any choice of $\beta_2$ without depending on the bounded noise assumption. ADOPT addresses the non-convergence issue of Adam by removing the current gradient from the second moment estimate and changing the order of the momentum update and the normalization by the second moment estimate. We also conduct intensive numerical experiments, and verify that our ADOPT achieves superior results compared to Adam and its variants across a wide range of tasks, including image classification, generative modeling, natural language processing, and deep reinforcement learning. The implementation is available at https://github.com/iShohei220/adopt.

## 1 Introduction

Stochastic optimization algorithms, such as stochastic gradient descent (SGD), play a central role in deep learning. In particular, adaptive gradient methods based on exponential moving averages, such as Adam [Kingma and Ba, 2014], are widely used in practice. Despite the empirical success, it is

38th Conference on Neural Information Processing Systems (NeurIPS 2024).

known that Adam does not converge in theory in general cases. For example, Reddi et al. [2018] show that Adam fails to converge to a correct solution in a simple example where the objective function at time $t$ is given as:

$$f_t(\theta) = \begin{cases} C\theta, & \text{for } t \bmod 3 = 1 \\ -\theta, & \text{otherwise,} \end{cases} \tag{1}$$

where $C > 2$ and $\theta \in [-1, 1]$. In this online optimization setting, Adam converges to a wrong solution (i.e., $\theta = 1$) instead of the true solution (i.e., $\theta = -1$) especially when the hyperparameter $\beta_2$ is set to a small value. There have been several attempts to fix the non-convergent behavior of Adam [Reddi et al., 2018, Zou et al., 2019]. For example, AMSGrad [Reddi et al., 2018] ensures the convergence for online convex optimization by making slight modifications to the Adam algorithm. Subsequent studies [Chen et al., 2019, Zhou et al., 2018] show that AMSGrad also converges to a stationary point for smooth nonconvex stochastic optimization problems. However, the convergence proofs rely on the assumption that the gradient noise is uniformly bounded. This assumption is stronger than the one used for the analysis of vanilla SGD [Ghadimi and Lan, 2013, Bertsekas and Tsitsiklis, 2000, Khaled and Richtárik, 2023], where the gradient *variance* is assumed to be uniformly bounded. In fact, the bounded noise assumption is often violated in practice. For example, when Gaussian noise is used in the gradient estimation (e.g., variational autoencoders [Kingma and Welling, 2014] and diffusion models [Ho et al., 2020, Song et al., 2021]), the stochastic gradient is no longer bounded.

Concurrently, Zhou et al. [2019] analyze the non-convergence of Adam in the problem described in Eq. (1) from the perspective of the correlation between the current gradient and the second moment estimate based on the exponential moving average. Specifically, they show that the non-convergence problem can be resolved by excluding the gradient of some recent steps from the calculation of the second moment estimate. Based on the analysis, they propose AdaShift, another variant of Adam. However, their theoretical analysis is limited to a single online convex problem described in Eq. (1), and the convergence of AdaShift for general nonconvex problems is unclear.

More recently, some works have demonstrated that Adam can converge by choosing $\beta_2$ in a problem-dependent manner [Shi et al., 2020, Zhang et al., 2022, Wang et al., 2022, Li et al., 2023, Wang et al., 2023]. However, tuning $\beta_2$ for each specific problem is troublesome; hence developing algorithms with the problem-independent convergence guarantee is still important to safely apply adaptive gradient methods to a wide range of machine learning problems.

In this paper, we propose an alternative approach to addressing the non-convergence problem of Adam without relying on the choice of $\beta_2$ or strong assumptions such as the bounded noise assumption. To derive our algorithm, we first examine the case without momentum, analyzing the convergence bound of RMSprop for general smooth nonconvex optimization problems. Through the analysis, we uncover the fundamental cause of non-convergence, which stems from the correlation between the second moment estimate and the current gradient. This finding aligns with the results demonstrated by Zhou et al. [2019] for online convex optimization. This correlation can be easily eliminated by excluding the current gradient from the second moment estimate.

Subsequently, we extend our findings to the case where momentum is incorporated, as in Adam, and discover that the Adam-style momentum also contributes to non-convergence. To address it, we propose to change the order of the momentum update and the normalization by the second moment estimate. With this small adjustment, we successfully eliminate the non-convergence problem of Adam without relying on a specific hyperparameter choice and the bounded noise assumption. We provide theoretical evidence demonstrating that our derived algorithm, named ADOPT, can achieve convergence with the optimal rate of $\mathcal{O}(1/\sqrt{T})$ for smooth nonconvex optimization.

In our experiments, we begin by assessing the performance of ADOPT in a toy example where Adam typically fails to converge depending on the choice of $\beta_2$. This toy example is an extension of the one presented in Eq. (1) by Reddi et al. [2018], but we consider a scenario where AMSGrad is also hard to converge due to the dependence on the bounded noise assumption. Our results demonstrate that ADOPT rapidly converges to the solution, while Adam fails to converge, and AMSGrad exhibits extremely slow convergence. Next, we conduct an experiment using a simple multi-layer perceptron on the MNIST classification task to evaluate the performance of ADOPT in nonconvex optimization. Our findings indicate that ADOPT outperforms existing adaptive gradient methods, including Adam, AMSGrad, and AdaShift. Finally, we evaluate the performance of ADOPT in various practical

applications, such as image classification of CIFAR-10 and ImageNet using ResNet [He et al., 2016] and SwinTransformer [Liu et al., 2021], training of deep generative models (NVAE), fine-tuning of language models (LLaMA), and deep reinforcement learning for continuous control. Our empirical results demonstrate that ADOPT achieves superior results over existing algorithms (e.g., Adam) in these practical applications.

## 2 Preliminary

### 2.1 Problem Definition

We consider the minimization of the objective function $f : \mathbb{R}^D \to \mathbb{R}$ with respect to the parameter $\boldsymbol{\theta} \in \mathbb{R}^D$. In this context, we focus on first-order stochastic optimization methods, where only the stochastic gradient $\boldsymbol{g}$ is accessible. As the objective $f$ can be nonconvex, the goal is to find a stationary point where $\nabla f(\boldsymbol{\theta}) = 0$ [Blair, 1985, Vavasis, 1995]. In order to analyze the convergence behavior of stochastic optimization algorithms, the following assumptions are commonly employed in the literature:

**Assumption 2.1.** *The objective function $f(\boldsymbol{\theta})$ is lower-bounded, i.e., $f(\boldsymbol{\theta}) \geq f_{\inf} > -\infty$ for all $\boldsymbol{\theta}$.*

**Assumption 2.2.** *The stochastic gradient $\boldsymbol{g}_t$ is an unbiased estimator of the objective $f(\boldsymbol{\theta}_{t-1})$, i.e., $\mathbb{E}[\boldsymbol{g}_t] = \nabla f(\boldsymbol{\theta}_{t-1})$ for all $t \geq 1$.*

**Assumption 2.3.** *The objective function is $L$-smooth on $\mathbb{R}^D$, i.e., there exists a constant $L > 0$ such that $\|\nabla f(\boldsymbol{x}) - \nabla f(\boldsymbol{y})\| \leq L\|\boldsymbol{x} - \boldsymbol{y}\|$ for all $\boldsymbol{x}, \boldsymbol{y} \in \mathbb{R}^D$.*

**Assumption 2.4.** *Variance of the stochastic gradient is uniformly bounded , i.e., there exists a constant $\sigma > 0$ such that $\mathbb{E}[\|\boldsymbol{g}_t - \nabla f(\boldsymbol{\theta}_{t-1})\|^2] \leq \sigma^2$.*

For the analysis of adaptive gradient methods (e.g., Adam and AdaGrad), many of previous works [Défossez et al., 2022, Li and Orabona, 2019, Ward et al., 2020, Zou et al., 2018] use a little stronger assumption instead of Assumption 2.4 for ease of proofs:

**Assumption 2.5.** *The stochastic gradient has a finite second moment, i.e., there exists a constant $G > 0$ such that $\mathbb{E}[\|\boldsymbol{g}_t\|^2] \leq G^2$.*

Assumption 2.5 requires that the true gradient $\nabla f$ is also uniformly bounded in addition to the variance of the stochastic gradient $\boldsymbol{g}$. Moreover, the convergence proof of AMSGrad tends to rely on an even stronger assumption as follows [Chen et al., 2019, Zhou et al., 2018].

**Assumption 2.6.** *The stochastic gradient is uniformly upper-bounded, i.e., there exists a constant $G > 0$ such that $\|\boldsymbol{g}_t\| \leq G$.*

In Assumption 2.6, the gradient noise $\xi_t := \boldsymbol{g}_t - \nabla f$ is assumed to be bounded almost surely in addition to the true graidient $\nabla f$. Note that when Assumption 2.6 holds, Assumption 2.5 is automatically satisfied; hence, Assumption 2.6 is a stronger assumption compared to Assumption 2.5. In this paper, we adopt Assumptions 2.1, 2.2, 2.3 and 2.5 for analysis, because one of our motivations is to address the omission of Assumption 2.6. In the analysis, we derive the upper bound of $\min_t\{\mathbb{E}[\|\nabla f(\boldsymbol{\theta}_t))\|^{4/3}]^{3/2}\}$ to investigate the convergence rate of the stochastic optimization algorithms, which is commonly performed in the literature [Défossez et al., 2022, Zou et al., 2019].

### 2.2 Review of Stochastic Optimization Algorithms for Nonconvex Objectives

The convergence of the vanilla SGD have been studied extensively in previous works. For smooth nonconvex functions, Ghadimi and Lan [2013] showed that SGD with a constant learning rate converges with an $\mathcal{O}(1/\sqrt{T})$ rate under Assumptions 2.1-2.4 by setting $\alpha_t = \alpha = \Theta(1/\sqrt{T})$, where $\alpha_t$ is a learning rate at the $t$-th step, and $T$ is a total number of parameter updates. This convergence rate is known to be minimax optimal up to a constant [Drori and Shamir, 2020]. For the diminishing learning rate scheme, the convergence bound of $\mathcal{O}(\log T/\sqrt{T})$ is well-known for $\alpha_t = \alpha/\sqrt{t}$ [Ghadimi and Lan, 2013]. Recently, Wang et al. [2021] have proved that SGD with $\alpha_t = \alpha/\sqrt{t}$ can also achieve the optimal rate $\mathcal{O}(1/\sqrt{T})$ by additionally assuming that the objective $f$ is upper-bounded.

While the vanilla SGD is still one of the most popular choices for stochastic optimization, adaptive gradient methods are dominantly used especially for deep learning. In adaptive gradient methods, the parameter $\boldsymbol{\theta}$ is updated additionally using the second moment estimate $\boldsymbol{v}_t$ in the following form:

$$\boldsymbol{\theta}_t = \boldsymbol{\theta}_{t-1} - \alpha_t \frac{\boldsymbol{g}_t}{\sqrt{\boldsymbol{v}_t + \epsilon^2}}, \tag{2}$$

where $\epsilon$ is a small positive constant. The division between vectors is applied in an element-wise manner, and the addition between a vector $\boldsymbol{a}$ and a scalar $b$ is defined as $(\boldsymbol{a} + b)_i \coloneqq a_i + b$. In AdaGrad [Duchi et al., 2011], $\boldsymbol{v}_t$ is defined as $\boldsymbol{v}_0 = \boldsymbol{0}$ and $\boldsymbol{v}_t = \boldsymbol{v}_{t-1} + \boldsymbol{g}_t \odot \boldsymbol{g}_t$. In RMSprop [Hinton et al., 2012], an exponential moving average is substituted for the simple summation, i.e., $\boldsymbol{v}_t = \beta_2 \boldsymbol{v}_{t-1} + (1 - \beta_2) \boldsymbol{g}_t \odot \boldsymbol{g}_t$, where $0 \le \beta_2 < 1$. Adam [Kingma and Ba, 2014] uses momentum in addition to the second moment estimate to accelerate the convergence as follows:

$$\boldsymbol{m}_t = \beta_1 \boldsymbol{m}_{t-1} + (1 - \beta_1) \boldsymbol{g}_t, \tag{3}$$

$$\boldsymbol{\theta}_t = \boldsymbol{\theta}_{t-1} - \alpha_t \frac{\boldsymbol{m}_t}{\sqrt{\boldsymbol{v}_t + \epsilon^2}}, \tag{4}$$

where $\boldsymbol{m}_0 = \boldsymbol{0}$. Here, we omit the bias correction technique used in the original paper for clarity. Unfortunately, RMSprop and Adam are not guaranteed to converge even in a simple convex optimization problem as demonstrated by Reddi et al. [2018], whereas AdaGrad with a constant learning rate is known to converge with an $\mathcal{O}(\log T/\sqrt{T})$ rate under Assupmtions 2.1-2.3 and 2.5 for smooth nonconvex cases [Li and Orabona, 2019, Ward et al., 2020, Zou et al., 2018, Chen et al., 2019, Défossez et al., 2022]. Although the convergence of Adam can be assured by choosing $\beta_2$ in a problem-dependent manner [Shi et al., 2020, Zhang et al., 2022, Wang et al., 2022, Li et al., 2023, Wang et al., 2023], it is difficult to know the proper choice of $\beta_2$ for each problem before training.

To fix the non-convergence of Adam without depending on $\beta_2$, some researchers have proposed variants of Adam. Reddi et al. [2018] proposed AMSGrad, which substitute $\hat{\boldsymbol{v}}_t$ for $\boldsymbol{v}$ in Eq. (3), where $\hat{\boldsymbol{v}}_0 = \boldsymbol{0}$ and $\hat{\boldsymbol{v}}_t = \max\{\hat{\boldsymbol{v}}_{t-1}, \boldsymbol{v}_t\}$. The idea behind AMSGrad is that the scaling factor $\sqrt{\hat{\boldsymbol{v}}_t + \epsilon^2}$ should be non-decreasing to ensure the convergence. After Reddi et al. [2018] originally proved the convergence of AMSGrad for online convex optimization, Chen et al. [2019] showed that AMSGrad with $\alpha_t = \alpha/\sqrt{t}$ converges with $\mathcal{O}(\log T/\sqrt{T})$ for nonconvex settings. Zhou et al. [2018] also analyzed the convergence of AMSGrad for nonconvex optimization, and derived the convergence rate of $\mathcal{O}(1/\sqrt{T})$ for a constant learning rate of $\alpha_t = \alpha = \Theta(1/\sqrt{T})$. However, their results depend on Assumption 2.6, which is often violated in practice. For example, variational autoencoders [Kingma and Welling, 2014] and diffusion models [Ho et al., 2020, Song et al., 2021] are typical examples in which Assumption 2.6 does not hold because they utilize unbounded Gaussian noise in the gradient estimation. The cause of requirement for Assumption 2.6 is the max operation in the definition of $\hat{\boldsymbol{v}}_t$. Since the max operation is convex, $\mathbb{E}[\hat{\boldsymbol{v}}_t] \le \max_t\{\mathbb{E}[\boldsymbol{v}_t]\}$ does not hold; hence Assumption 2.6 is required to upper-bound $\mathbb{E}[\hat{\boldsymbol{v}}_t]$ in their proofs.

Zhou et al. [2019] also tried to fix the non-convergent behavior of Adam. Their proposed AdaShift uses $\boldsymbol{v}_{t-n}$ instead of $\boldsymbol{v}_t$ for the second moment estimate, and calculate the momentum using the latest $n$ gradients as follows:

$$\boldsymbol{m}_t = \frac{\sum_{k=0}^{n-1} \beta_1^k \boldsymbol{g}_{t-k}}{\sum_{k=0}^{n-1} \beta_1^k}, \tag{5}$$

$$\boldsymbol{\theta}_t = \boldsymbol{\theta}_{t-1} - \alpha_t \frac{\boldsymbol{m}_t}{\sqrt{\boldsymbol{v}_{t-n} + \epsilon^2}}. \tag{6}$$

In the original paper, some additional techniques (e.g., the block-wise adaptive learning rate) are used, but we omit them for clarity here. Though they give theoretical analysis for a single online convex example, any convergence bounds are not provided for nonconvex cases. More detailed discussion on existing analyses is provided in Appendix A.

## 3 Analysis: Cause of Non-convergence of Adam and How to Fix It

In this section, to derive an algorithm that can converge with any $\beta_2$ without Assumption 2.6, we analyze the cause of non-convergence of Adam, and discuss how it can be eliminated. To start from a simple case, we first analyze the case without momentum. Subsequently, we extend it to the case with momentum and provide a way to fix the convergence issue of Adam.

## 3.1 Case without Momentum

We first analyze the convergence of RMSprop, which corresponds to the no-momentum case of Adam when we omit the bias correction. For RMSprop, we derive the following convergence bound.

**Theorem 3.1.** *Under Assumptions 2.1-2.3 and 2.5, the following holds for the RMSprop with a constant learning rate $\alpha_t = \alpha$:*

$$\min_{t=1,\ldots,T} \left\{ \mathbb{E}\left[ \|\nabla f(\boldsymbol{\theta}_{t-1}))\|^{4/3} \right]^{3/2} \right\} \leq C_1 \left( \frac{f_0 - f_{\inf}}{\alpha T} + \frac{C_2}{T} \log\left(1 + \frac{G^2}{\epsilon^2}\right) - C_2 \log\beta_2 \right), \quad (7)$$

*where $C_1 = 2\sqrt{G^2 + \epsilon^2}$, $C_2 = \frac{\alpha DL}{2(1-\beta_2)} + \frac{2DG}{\sqrt{1-\beta_2}}$, and $f_0 = f(\boldsymbol{\theta}_0)$.*

*Sketch of proof.* By Assumption 2.3, the following holds:

$$\mathbb{E}\left[ f(\boldsymbol{\theta}_t) \right] \leq \mathbb{E}\left[ f(\boldsymbol{\theta}_{t-1}) + \frac{\alpha^2 L}{2} \left\| \frac{\boldsymbol{g}_t}{\sqrt{\boldsymbol{v}_t + \epsilon^2}} \right\|^2 - \alpha \nabla f(\boldsymbol{\theta}_{t-1})^\top \left( \frac{\boldsymbol{g}_t}{\sqrt{\boldsymbol{v}_t + \epsilon^2}} \right) \right] \quad (8)$$

Applying Lemmas G.4 and G.6 in the appendix to this, the following inequality is derived:

$$\mathbb{E}\left[ f(\boldsymbol{\theta}_t) \right]$$
$$\leq \mathbb{E}\left[ f(\boldsymbol{\theta}_{t-1}) + \left( \frac{\alpha^2 L}{2} + 2\alpha G\sqrt{1-\beta_2} \right) \left\| \frac{\boldsymbol{g}_t}{\sqrt{\boldsymbol{v}_t + \epsilon^2}} \right\|^2 - \frac{\alpha}{2} \nabla f(\boldsymbol{\theta}_{t-1})^\top \left( \frac{\boldsymbol{g}_t}{\sqrt{\tilde{\boldsymbol{v}}_t + \epsilon^2}} \right) \right] \quad (9)$$

$$\leq \mathbb{E}\left[ f(\boldsymbol{\theta}_{t-1}) + \left( \frac{\alpha^2 L}{2} + 2\alpha G\sqrt{1-\beta_2} \right) \left\| \frac{\boldsymbol{g}_t}{\sqrt{\boldsymbol{v}_t + \epsilon^2}} \right\|^2 \right] - \frac{\alpha}{2} \frac{\mathbb{E}\left[ \|\nabla f(\boldsymbol{\theta}_{t-1})\|^{4/3} \right]^{3/2}}{\sqrt{\left(1 - \beta_2^T\right)G^2 + \epsilon^2}}, \quad (10)$$

where $\tilde{\boldsymbol{v}}_t = \beta_2 \boldsymbol{v}_{t-1} + (1-\beta_2)\mathbb{E}[\boldsymbol{g}_t \odot \boldsymbol{g}_t]$. Telescoping this for $t = 1, \ldots, T$ and rearranging the terms, we have

$$\sum_{t=1}^{T} \mathbb{E}\left[ \|\nabla f(\boldsymbol{\theta}_{t-1})\|^{4/3} \right]^{3/2} \leq C_1 \left( \frac{f(\boldsymbol{\theta}_0) - f_{\inf}}{\alpha} + C_2 \log\left( \frac{G^2 + \epsilon^2}{\beta_2^T \epsilon^2} \right) \right), \quad (11)$$

where the last inequality holds due to Assumption 2.1 and Lemma G.5. Therefore, the bound in Eq. (7) is derived using $\min_{t=1,\ldots,T}\{\mathbb{E}[\|\nabla f(\boldsymbol{\theta}_{t-1}))\|^{4/3}]^{3/2}\} \leq \sum_{t=1}^{T} \mathbb{E}[\|\nabla f(\boldsymbol{\theta}_{t-1})\|^{4/3}]^{3/2}/T.$ $\quad\square$

A detailed proof is provided in the appendix. When the learning rate $\alpha$ is chosen so that $\alpha = \Theta(1/\sqrt{T})$, the first and second terms on the right hand side of Eq. (7) converge with $\mathcal{O}(1/\sqrt{T})$ and $\mathcal{O}(1/T)$ rates, respectively. However, the last term includes a constant factor in terms of $T$, which represents the non-convergent behavior of RMSprop in the smooth nonconvex setting. More precisely, RMSprop is guaranteed to converge only to a bounded region around a stationary point, and the size of the bounded region depends on the hyperparameter $\beta_2$ and the problem-dependent factors $D$, $G$, and $L$. Therefore, we need to choose $\beta_2$ dependently on each problem to make the bounded region adequately small. Since $\lim_{\beta_2 \to 1} \log\beta_2/\sqrt{1-\beta_2} = 0$, the size of the bounded region can be made small by setting $\beta_2$ to a value close to 1, which aligns with practical observations. However, how close to 1 it should be relies on the problem-dependent factors, which cannot be observed in advance. This result is consistent with recent results of convergence analyses of Adam and RMSprop [Shi et al., 2020, Zhang et al., 2022].

As can be seen from Eqs. (8) and (9), the constant term in Eq. (7) is derived from the last term of Eq. (8). Because $\boldsymbol{g}_t$ and $\boldsymbol{v}_t$ are not statistically independent, this term is first decomposed as in Eq. (9). After the decomposition, $\boldsymbol{g}_t$ and $\tilde{\boldsymbol{v}}_t$ is now conditionally independent given $\boldsymbol{g}_0, \ldots, \boldsymbol{g}_{t-1}$, so Eq. (10) is derived using the following fact:

$$\mathbb{E}\left[ \frac{\boldsymbol{g}_t}{\sqrt{\tilde{\boldsymbol{v}}_t + \epsilon^2}} \right] = \mathbb{E}\left[ \frac{\nabla f(\boldsymbol{\theta}_{t-1})}{\sqrt{\tilde{\boldsymbol{v}}_t + \epsilon^2}} \right]. \quad (12)$$

This indicates that, if the second moment estimate $\boldsymbol{v}_t$ is designed to be conditionally independent to $\boldsymbol{g}_t$, the constant term in the convergence bound will be removed, because the second term of Eq. (8)

---

**Algorithm 1** ADOPT algorithm

---

**Require:** Learning rate $\{\alpha_t\}$, initial parameter $\boldsymbol{\theta}_0$
**Require:** Exponential decay rate $0 \leq \beta_1 < 1, 0 \leq \beta_2 \leq 1$, small constant $\epsilon > 0$
   $\boldsymbol{v}_0 \leftarrow \boldsymbol{g}_0 \odot \boldsymbol{g}_0, \boldsymbol{m}_1 \leftarrow \boldsymbol{g}_1 / \max\left\{\sqrt{\boldsymbol{v}_0}, \epsilon\right\}$
   **for** $t = 1$ to $T$ **do**
      $\boldsymbol{\theta}_t \leftarrow \boldsymbol{\theta}_{t-1} - \alpha_t \boldsymbol{m}_t$
      $\boldsymbol{v}_t \leftarrow \beta_2 \cdot \boldsymbol{v}_{t-1} + (1 - \beta_2)\, \boldsymbol{g}_t \odot \boldsymbol{g}_t$
      $\boldsymbol{m}_{t+1} \leftarrow \beta_1 \cdot \boldsymbol{m}_t + (1 - \beta_1)\, \frac{\boldsymbol{g}_{t+1}}{\max\{\sqrt{\boldsymbol{v}_t}, \epsilon\}}$
   **end for**
   **return** $\{\boldsymbol{\theta}_t\}_{t=1}^{T}$

---

can be directly lower-bounded without the decomposition. A simple way to achieve the conditional independence is to substitute $\boldsymbol{v}_{t-1}$ for $\boldsymbol{v}_t$ as a second moment estimate, because $\boldsymbol{v}_{t-1}$ does not have information about $\boldsymbol{g}_t$. This solution is similar to AdaShift, in which $\boldsymbol{v}_{t-n}$ is substituted for $\boldsymbol{v}_t$ as described in Eq. (5). In fact, the modified version of RMSprop is identical to AdaShift with $n = 1$ and $\beta_1 = 0$ except for the additional techniques (e.g., the block-wise adaptive learning rate).

## 3.2 Case with Momentum

As we have described, RMSprop can be modified to be convergent by removing the current gradient $\boldsymbol{g}_t$ from the second moment estimate $\boldsymbol{v}_t$. However, when we combine adaptive gradient methods with momentum like Adam, the convergence analysis becomes more complicated. Unfortunately, when Adam-style momentum in Eq. (3) is applied, the algorithm does not converge in general even when using $\boldsymbol{v}_{t-1}$ as a second moment estimate instead of $\boldsymbol{v}_t$. This is because the momentum $\boldsymbol{m}_t$ contains all history of the past gradients $\boldsymbol{g}_0, \ldots, \boldsymbol{g}_t$; hence the second moment estimate always correlates with $\boldsymbol{m}_t$. AdaShift prevents this problem by calculating the momentum $\boldsymbol{m}_t$ only using the latest $n$ gradients as described in Eq. (5). In that case, the momentum $\boldsymbol{m}_t$ and the second moment estimate $\boldsymbol{v}_{t-n}$ are conditionally independent, so the convergence can be retained. However, this approach has a trade-off in the choice of $n$. When $n$ is small, $\boldsymbol{m}_t$ has little information about the past gradients; when $n$ is large, $\boldsymbol{v}_{t-n}$ only has access to the gradient information in the distant past.

To remove this trade-off, instead of truncating the momentum to the latest $n$ steps, we propose to use momentum of the following form:

$$\boldsymbol{m}_t = \beta_1 \boldsymbol{m}_{t-1} + (1 - \beta_1)\, \frac{\boldsymbol{g}_t}{\sqrt{\boldsymbol{v}_{t-1} + \epsilon^2}}, \tag{13}$$

$$\boldsymbol{\theta}_t = \boldsymbol{\theta}_{t-1} - \alpha_t \boldsymbol{m}_t. \tag{14}$$

The main difference to the Adam-style momentum in Eq. (3) is the order of update of $\boldsymbol{m}_t$ and the normalization by $\sqrt{\boldsymbol{v}_{t-1} + \epsilon^2}$. In Eq. (3), the normalization is performed after the update of $\boldsymbol{m}_t$, whereas in Eq. (13), the normalization is first applied to the current gradient $\boldsymbol{g}_t$ in advance to the update of $\boldsymbol{m}_t$. In this case, the second moment estimate $\boldsymbol{v}_{t-1}$ is only used to normalize the current gradient $\boldsymbol{g}_t$, so the convergence can be guaranteed. A more detailed convergence analysis is provided in Section 4.

# 4 Method: Adaptive Gradient Method with the Optimal Convergence Rate

Based on the analysis in the previous section, we propose a new adaptive gradient method named ADOPT (*ADaptive gradient method with the OPTimal convergence rate*). The entire procedure is summarized in Algorithm 4. For a simple discription, we place the update of $\boldsymbol{m}$ after the parameter update in Algorithm 4, but it is equivalent to Eqs. (13) and (14) except that $\max\left\{\sqrt{\boldsymbol{v}}, \epsilon\right\}$ is substitued for $\sqrt{\boldsymbol{v} + \epsilon^2}$. The substitition is applied because we find that it contributes to slightly better performance in practice. We provide an equivalent expression of Algorithm 4 in Algorithm C in the appendix, which is closer to a practical implementation. By this modification, ADOPT can converge with the optimal rate for smooth nonconvex optimization as follows:

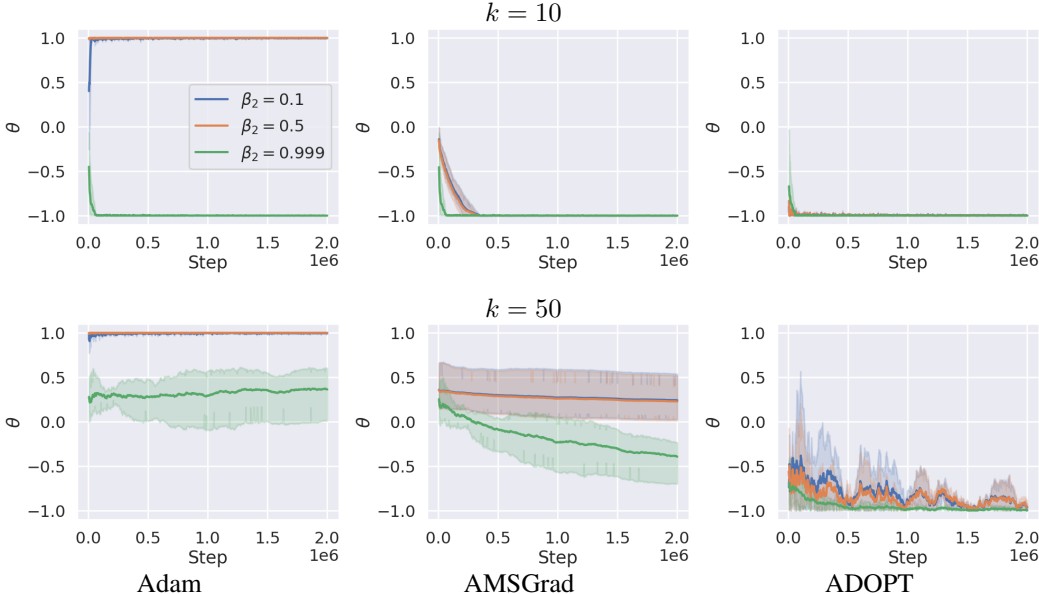

Figure 1: Performance comparison between Adam, AMSGrad and ADOPT in a simple univariate convex optimization problem. The plots show transitions of the parameter value, which should converge to the solution $\theta = -1$.

**Theorem 4.1.** *Under Assumptions 2.1-2.3 and 2.5, the following holds for the ADOPT algorithm with a constant learning rate $\alpha_t = \alpha = \Theta\left(1/\sqrt{T}\right)$:*

$$\min_{t=1,\ldots,T}\left\{\mathbb{E}\left[\|\nabla f(\boldsymbol{\theta}_{t-1}))\|^{4/3}\right]^{3/2}\right\} \leq \mathcal{O}\left(1/\sqrt{T}\right), \tag{15}$$

The detailed proof and related lemmas are provided in the appendix. We also provide the convergence bound for the case of diminishing learning rate (i.e., $\alpha_t = \alpha/\sqrt{t}$) in the appendix, which is closer to practical situations. In that case, ADOPT also converges with the optimal rate.

## 5 Experiments

In the experiments, we first validate our ADOPT algorithm using a simple toy example in which Adam is known to fail to converge, and confirm our theoretical findings through numerical simulation. Secondly, we run an experiment of training a simple multi-layer perceptron (MLP) for the MNIST dataset to verify the effectiveness of our ADOPT for nonconvex optimization problems. Finally, we evaluate our ADOPT in a wide range of practical applications, including image classification, natural language processing (NLP) tasks, generative modeling, and deep reinforcement learning. Detailed experimental settings are described in the appendix.

**Toy problem:** We consider a convex optimization problem with an objective $f(\theta) = \theta$ for $\theta \in [-1, 1]$. It is obvious that a solution for the problem is $\theta = -1$. Through the optimization, we only have access to the stochastic objective $f_t$ as follows:

$$f_t(\theta) = \begin{cases} k^2\theta, & \text{with probability } 1/k \\ -k\theta, & \text{with probability } 1 - 1/k \end{cases}, \tag{16}$$

where $k \geq 1$. Because $\mathbb{E}[f_t(\theta)] = f(\theta)$ holds, the stochastic gradient $g_t = \nabla f_t(\theta)$ is an unbiased estimator of the true gradient $\nabla f$ regardless of the choice of $k$, satisfying Assumption 2.2. This problem is equivalent, except for scaling, to the stochastic optimization version of Eq. (1) provided by Reddi et al. [2018] as a case where Adam fails to converge. In this setting, the constant $k$ controls the magnitude of gradient noise. When $k = 1$, it corresponds to the noiseless case where $f_t = f$ with probability 1. As $k$ gets large, stochastic gradient becomes noisy, making $G$ in Assumptions

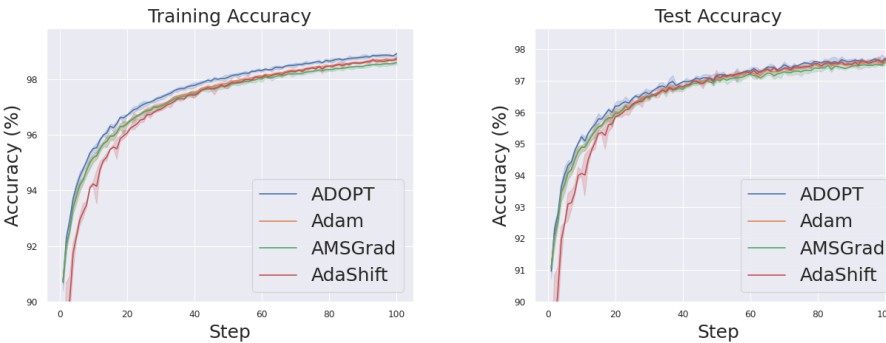

Figure 2: Accuracy for training data (left) and test data(right) in MNIST classification. The error bars show the 95% confidence intervals of three trials.

2.5 and 2.6 large. Therefore, the optimization will be more difficult when $k$ becomes larger. In the experiment, we set $k = 10$ or $50$, and compare the robustness of Adam, AMSGrad, and ADOPT for various hyperparameter settings by changing $\beta_2$ from $0.1 \sim 0.999$. We set $\beta_1 = 0.9$ for all the algorithms, which is a common choice in practice. We set the learning rate to $\alpha_t = 0.01/\sqrt{1 + 0.01t}$.

The result is shown in Figure 1. It can be seen that, when $k = 10$, Adam fails to converge except for $\beta_2 = 0.999$ while AMSGrad and ADOPT rapidly converge to the correct solution, i.e., $\theta = -1$, with any $\beta_2$. In a more extreme case where $k = 50$, Adam fails to converge even with $\beta_2 = 0.999$. This aligns with Theorem 3.1, since, when the gradient noise is large (i.e., $G$ is large), the bounded region of the convergence bound also gets large, leading to divergence of Adam. Moreover, when $k = 50$, it is observed that the convergence of AMSGrad also becomes much slower than ADOPT. In fact, this phenomenon is also consistent with theory. In this problem setting, the second moment $\mathbb{E}[g_t^2]$ is $\mathcal{O}(k^3)$, while the squared norm of the stochastic gradient $g_t^2$ is $\mathcal{O}(k^4)$. Since the convergence bound of AMSGrad depends on the uniform bound of the stochastic gradient in Assumption 2.6, instead of the second moment in Assumption 2.5, its convergence also deteriorates with the order of $g_t^2$. Compared to AMSGrad, ADOPT only depends on the second moment bound for its convergence, so it converges much faster than AMSGrad even in such an extreme setting.

We also perform ablation study on how the two algorithmic changes from Adam to ADOPT affect the convergence. The differences between Adam and ADOPT are (1) decorrelation between the second moment estimate and the current gradient, and (2) change of order of momentum update and normalization by the second moment estimate. In this experiment, we remove each algorithmic change from ADOPT, and compare the result in the toy example. We set $k = 50$, and $(\beta_1, \beta_2) = (0.9, 0.999)$, since it is a common hyperparameter choice. The result is shown in Figure 3. It can be observed that ADOPT fails to converge with the exception of either algorithmic change. Therefore, applying both changes is essential to overcome the non-convergence of Adam, which also aligns with theory. These results correspond to the theoretical findings, showing the superiority of ADOPT to Adam and AMSGrad in terms of the convergence speed and its robustness to hyperparameter choices.

**MNIST classification:** To investigate the effectiveness of ADOPT on nonconvex optimization, we train nonlinear neural networks for MNIST classification tasks, and compare the performance between ADOPT and existing optimization algorithms, such as Adam, AMSGrad and AdaShift. In this experiment, we use a simple MLP with a single hidden layer, and the number of hidden units is set to 784. We set the learning rate to $\alpha_t = \alpha/\sqrt{t}$, and $\alpha$ is tuned in the range of $\{1, 10^{-1}, 10^{-2}, 10^{-3}\}$. We apply weight decay of $1 \times 10^{-4}$ to prevent over-fitting, and run 10K iterations of parameter updates. Figure 2 shows the learning curves of training and test accuracy. We observe our ADOPT performs slightly better than the others in terms of the convergence speed and the final performance.

**Image classification:** As a more practical application, we conduct experiments of image classification using real-world image datasets. We first compare ADOPT and Adam in the classification task of the CIFAR-10 dataset using ResNet-18 [He et al., 2016], a widely-used convolutional neural network. We conduct a similar hyperparameter search to the case of MNIST classification. A detailed experimental setting is provided in the appendix. The learning curves of test accuracy are visualized in Figure 4. It can be observed that ADOPT converges a little faster than Adam.

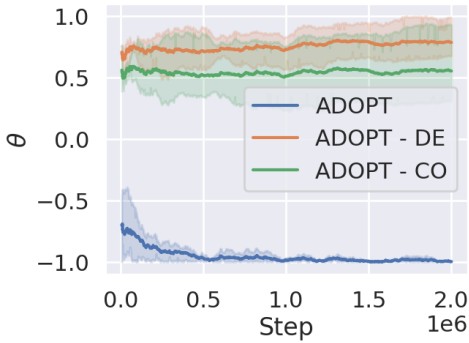

Figure 3: Ablation study of algorithmic changes between Adam and ADOPT. "DE" and CO denote "decorrelation" and "change of order", respectively.

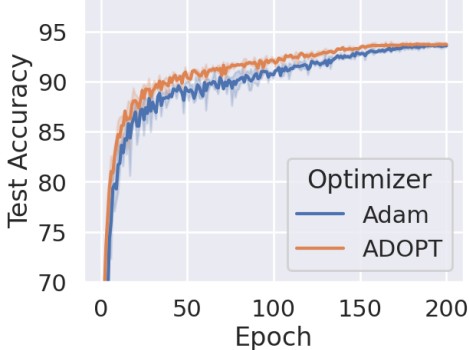

Figure 4: Learning curves of test accuracy for CIFAR-10 classification by ResNet-18 trained with Adam and ADOPT.

Table 1: Top-1 accuracy (%) for ImageNet classification by SwinTransformer.

| Epoch | 200 | 300 |
|---|---|---|
| AdamW | $79.29 \pm 0.05$ | $81.26 \pm 0.04$ |
| AMSGrad | $78.91 \pm 0.03$ | $81.17 \pm 0.03$ |
| ADOPT | $\mathbf{79.62} \pm 0.03$ | $\mathbf{81.50} \pm 0.04$ |

Table 2: Negative log-likelihood of NVAEs for MNIST density estimation. Lower is better.

| Epoch | 200 | 300 |
|---|---|---|
| Adamax | $80.19 \pm 0.08$ | $79.41 \pm 0.07$ |
| ADOPT | $\mathbf{79.02} \pm 0.10$ | $\mathbf{78.88} \pm 0.09$ |

To confirm that our ADOPT works well for modern neural network architectures based on Transformers [Vaswani et al., 2017], we perform an experiment of ImageNet classification using SwinTransformer [Liu et al., 2021]. We follow the official training recipe of Swin Transformer-tiny provided by Torchvision [Paszke et al., 2019a], and fix the training settings except for the optimizer choice. We use AdamW [Loshchilov and Hutter, 2019] as a baseline because it is set as the default official optimizer. We also compare with AMSGrad as another way to fix the non-convergence issue of Adam. Since AdamW uses decoupled weight decay, we also apply it to the other optimizers for fair comparison. We report the top-1 accuracy at 200 and 300 epochs in Tables 1. We observe that ADOPT outperforms AdamW and AMSGrad throughout the training in terms of the test accuracy, demonstrating the effectiveness of ADOPT for this setting.

**Generative modeling:** We train NVAE [Vahdat and Kautz, 2020] for MNIST using our ADOPT. In the official implementation of NVAE, Adamax [Kingma and Ba, 2014], an infinite-norm variant of Adam, is used as an optimizer, so we use Adamax as a baseline method. We use the exactly the same setting of the official implementation except that the learning rate for ADOPT is set to $2 \times 10^{-4}$ since the default value $0.01$ is too large for ADOPT. We report the negative log-likelihood for test data on Table 2. It is observed that the model trained with ADOPT shows the better likelihood.

**Pretraining of large language models:** We run a pre-training of GPT-2 [Radford et al., 2019] using the nanoGPT [Karpathy, 2022] code base to compare Adam and ADOPT. We use OpenWeb-Text [Gokaslan and Cohen, 2019] as the training data. Experimental setup conforms to the default settings of nanoGPT except for the selection of the optimizer. We also test a case in which the total batch size was changed from 480 to 96, as a setting where the gradient noise becomes larger. The results are summarized in Figure 5. The most notable finding is that in the small batch size case, Adam causes loss spikes in the early stages of training and fails to converge, while ADOPT is always able to train stably. This is consistent with Adam's theory of non-convergence. As the gradient noise increases, $G$ in Theorem 3.1 also increases, and the constant term in Adam's convergence bounds becomes non-negligible especially when using a large-scale dataset like OpenWebText. As a result, Adam is more likely to fail to train in such cases. Our ADOPT, on the other hand, does not suffer from this problem because it can always guarantee convergence. We also observed that both Adam and ADOPT work well when the batch size is large, but even in this case, ADOPT performs slightly better.

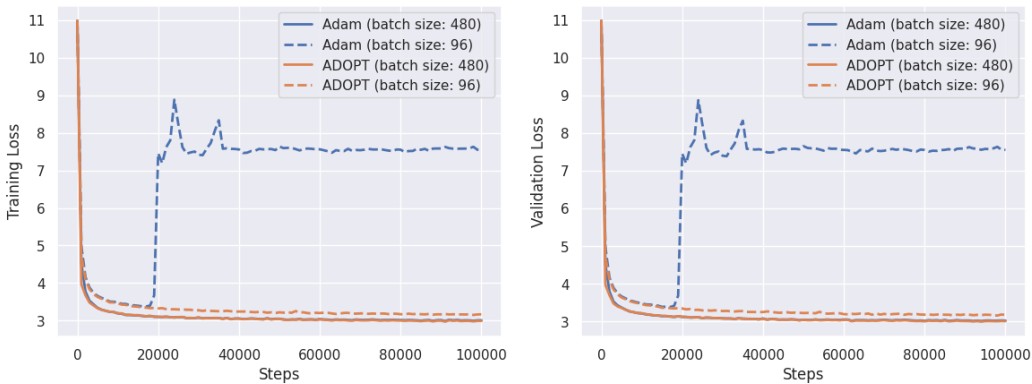

Figure 5: Learning curves of GPT-2 pretraining for training set (left) and validation set (right).

**Finetuning of large language models:** We finetune the pretrained LLaMA-7B on 52K instruction-following data provided by Stanford Alpaca and compare the performance between the default optimizer (Adam) and our ADOPT under the exactly same experimental setting. For evaluation, we use Multi-task Language Understanding (MMLU) Benchmark [Hendrycks et al., 2021], which is widely used to assess the performance of large language models. The MMLU score for LLaMA-7B without finetuning is 35.1. After fine-tuned via instruction-following using the baseline implementation with Adam, the score improves to 41.2. When we substitute ADOPT for Adam, the score even improves to 42.13. The detailed score comparison for each task is summarized in Figure 7 in the appendix. Other experimental results, including deep RL experiments, and detailed experimental settings are also provided in the appendix.

## 6 Conclusion

In this paper, we demystified the fundamental cause of divergence of adaptive gradient methods based on the exponential moving average, such as Adam and RMSprop, in general smooth nonconvex optimization problems, and demonstrate a way to fix the issue, proposing a new optimizer named ADOPT. Not only does ADOPT converge with the optimal rate without depending on a hyperparameter choice in theory, but ADOPT demonstrates better performance in a wide range of pracital applications.

We expect that this work will serve as a bridge between theory and practice in the research of adaptive gradient methods. Since ADOPT can be safely applied to many machine learning problems without careful tuning of hyperparameters, it can be expected to improve the training stability and the model performance in practice by substituting it for the existing adaptive gradient methods (e.g., Adam).

One of the limitations of our analysis is that it still relies on the assumption that the second moment of stochastic gradient is uniformly bounded (i.e., Assumption 2.5). Although this assumption is weaker than the bounded stochastic gradient assumption (i.e., Assumption 2.6), it would be more desirable to relax it to the bounded *variance* assumption (i.e., Assumption 2.4), which is often adopted in the analysis of the vanilla SGD [Ghadimi and Lan, 2013]. For Adam, a recent work by Wang et al. [2023] have derived a problem-dependent convergence bound which achieves the $\mathcal{O}(1/\sqrt{T})$ rate without Assumption 2.5. Their proof techniques may help to relax our assumptions in the proof of Theorem 4.1, which we leave as future work.

From a broader perspective, adaptive gradient methods like Adam have been widely used even for the training of large-scale foundation models (e.g., large language models). Although such models can be useful for people, their negative aspects, such as concerns about copyright infringement, are not negligible. Researchers needs to deeply recognize and understand such social impacts of machine learning algorithms.

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

# A  Detailed Relationships to Existing Analyses

In this section, we discuss the relationships between our analysis and existing ones on the convergence of Adam-like optimizers in smooth nonconvex optimization problems. Tables 3 and 4 are a summary of comparisons between them in terms of their problem settings and derived convergence rates.

**Zhang et al. [2022]** focus on convergence of Adam in the finite sum problem, where the objective has a following form:

$$f\left(\boldsymbol{\theta}\right) = \sum_{i=1}^{n} f_i\left(\boldsymbol{\theta}\right). \tag{17}$$

$f_i$ is, for example, a loss function for $i$-th training sample. Although many deep learning problems can be formulated as a finite sum problem, training of the variational autoencoders (VAEs) or diffusion models is out of the finite-sum problem, since their objective is formulated as an infinite sum (i.e., an expectation over continuous variables). Moreover, they assume the stochastic gradient $\boldsymbol{g}$ is $L$-Lipschitz, whereas we only assume true gradient $\nabla f$ is $L$-Lipschitz. They also assume a growth condition as follows:

$$\mathbb{E}\left[\|\boldsymbol{g}_t\|^2\right] \leq G_0^2 + G_1^2 \|\nabla f\left(\boldsymbol{\theta}_{t-1}\right)\|^2. \tag{18}$$

This growth condition is weaker than our Assumption 2.5. Assumption 2.5 is a special case of the growth condition where $G_1 = 0$. Their derived convergence rate has a constant factor of $\mathcal{O}(G_0)$; hence the strong growth condition (i.e., $G_0 = 0$) is required to assure convergence. Moreover, to assure convergence, one needs to choose sufficiently large $\beta_2$, which has to be tuned in a problem-dependent manner.

**Wang et al. [2022]** also focus on convergence of Adam in the finite sum problem, but they relax the $L$-Lipschitz condition on $\boldsymbol{g}$ to the $(L_0, L_1)$-Lipschitz condition. They also assume the growth condition in Eq. (18), and their convergence rate has the same order with Zhang et al. [2022], so it still requires the strong growth condition (i.e., $G_0 = 0$) to assure convergence. The condition of $\beta_2$ is also similar to Zhang et al. [2022].

**Li et al. [2023]** consider Adam's convergence on general smooth nonconvex problems. Similar to Wang et al. [2022], they use $(L_0, L_\rho)$-Lipschitz condition on the true gradient $\nabla f$. They also assume that the gradient noise is almost surely bounded:

$$\|\boldsymbol{g} - \nabla f\| \leq \sigma \tag{19}$$

The relationship between this assumption and our Assumption 2.5 is a little complicated. Assumption 2.5 is equivalent to a combination of Assumption 2.4 and the following assumption:

**Assumption A.1.** *The true gradient is uniformly bounded, i.e., there exist constants $G$ and $\sigma$ such that* $\|\nabla f\left(\boldsymbol{\theta}\right)\|^2 \leq G^2 - \sigma^2$ *and* $0 < \sigma \leq G$.

The bounded noise assumption of Eq. (19) is strictly stronger than Assumption 2.4, but they do not assume the bounded true gradient (i.e., Assumption A.1). The bounded noise assumption is often violated in practice (e.g., training of VAEs), because the gradient is often estimated using unbounded noise (i.e., Gaussian noise). Their convergence rate $\mathcal{O}(1/\sqrt{T})$ is better than Zhang et al. [2022] and Wang et al. [2022], while it still requires constraints on the hyperparameters, which have to be chosen in a problem-dependent manner.

**Défossez et al. [2022]** analyzes the convergence of Adam under exactly the same assumptions with ours, and they derive the $\mathcal{O}(\log T/\sqrt{T})$ rate, which is worse than our ADOPT's convergence rate. Moereover, to assure the convergence, $\beta_2$ has to be chosen dependently on the total number of iterations $T$.

**Wang et al. [2023]** analyzes the convergence of Adam under Assumptions 2.1-2.4, and they derive the $\mathcal{O}(1/\sqrt{T})$ rate. However, to assure the convergence, $\beta_2$ has to be chosen dependently on the total number of iterations $T$ as in Défossez et al. [2022].

**Chen et al. [2019]** and **Zhou et al. [2018]** analyze the convergence of AMSGrad for general smooth nonconvex problems, and derive the convergence rate of $\mathcal{O}(\log T/\sqrt{T})$ and $\mathcal{O}(1/\sqrt{T})$, respectively. However, to guarantee the convergence, the stochastic gradient $\boldsymbol{g}$ has to be bounded almost surely

| | Algorithm | Problem | Smoothness | Gradient Growth |
|---|---|---|---|---|
| Zhang et al. [2022] | Adam | Finite sum | $L$-Lipschitz $\boldsymbol{g}$ | $\mathbb{E}[\|g\|^2] \leq G_0^2 + G_1^2\|\nabla f\|^2$ |
| Wang et al. [2022] | Adam | Finite sum | $(L_0, L_1)$-Lipschitz $\boldsymbol{g}$ | $\mathbb{E}[\|g\|^2] \leq G_0^2 + G_1^2\|\nabla f\|^2$ |
| Li et al. [2023] | Adam | General | $(L_0, L_\rho)$-Lipschitz $\nabla f$ | $\|\boldsymbol{g} - \nabla f\| \leq \sigma$ |
| Défossez et al. [2022] | Adam | General | $L$-Lipschitz $\nabla f$ | $\mathbb{E}[\|g\|^2] \leq G^2$ |
| Wang et al. [2023] | Adam | General | $L$-Lipschitz $\nabla f$ | $\mathbb{E}[\|\boldsymbol{g} - \nabla f\|^2] \leq G^2$ |
| Chen et al. [2019] | AMSGrad | General | $L$-Lipschitz $\nabla f$ | $\|\boldsymbol{g}\| \leq G$ |
| Zhou et al. [2018] | AMSGrad | General | $L$-Lipschitz $\nabla f$ | $\|\boldsymbol{g}\| \leq G$ |
| Ours | ADOPT | General | $L$-Lipschitz $\nabla f$ | $\mathbb{E}[\|\boldsymbol{g}\|^2] \leq G^2$ |

Table 3: Comparison of the problem settings between our analysis and other existing works.

| | Constraints | Convergence |
|---|---|---|
| Zhang et al. [2022] | $\beta_1 < \sqrt{\beta_2}, \beta_2 \geq \gamma(n)$ | $\mathcal{O}(\log T/\sqrt{T}) + \mathcal{O}(G_0)$ |
| Wang et al. [2022] | $\beta_1 < \sqrt{\beta_2}, \delta(\beta_2) = \mathcal{O}(1/G_1)$ | $\mathcal{O}(\log T/\sqrt{T}) + \mathcal{O}(G_0)$ |
| Li et al. [2023] | $\beta_1 < \sqrt{\beta_2}, \beta_1 \leq c(L_0, L_\rho, G)$ | $\mathcal{O}(1/\sqrt{T})$ |
| Défossez et al. [2022] | $\beta_1 < \sqrt{\beta_2}, 1 - \beta_2 = \Theta(1/T)$ | $\mathcal{O}(\log T/\sqrt{T})$ |
| Wang et al. [2023] | $\beta_1 \leq \sqrt{\beta_2} - 8\left(1 - \beta_2\right)\beta_2^{-2}, 1 - \beta_2 = \Theta(1/T)$ | $\mathcal{O}(1/\sqrt{T})$ |
| Chen et al. [2019] | $\beta_1 < \sqrt{\beta_2}$ | $\mathcal{O}(\log T/\sqrt{T})$ |
| Zhou et al. [2018] | $\beta_1 < \sqrt{\beta_2}$ | $\mathcal{O}(1/\sqrt{T})$ |
| Ours | - | $\mathcal{O}(1/\sqrt{T})$ |

Table 4: Comparison of the convergence rate and imposed constraints on the hyperparameters between our analysis and other existing works. Please refer to the original papers for the definitions of $\gamma$ and $c$.

(Assumption 2.6), which is often violated in practice. In addition, the hyperparameter $\beta_1$ and $\beta_2$ should be chosen satisfying $\beta_1 < \sqrt{\beta_2}$. This constraint is relatively minor compared to the constraint imposed in the analyses of Adam, since it can be satisfied in a problem-independent manner.

## B  With-Replacement vs. Without-Replacement

In the optimization of finite-sum problems, practitioners often use *without-replacement sampling*, which is also known as *random shuffling*, to obtain stochastic gradient. In this case, the stochastic gradient has a small bias due to the lack of replacement, so Assumption 2.2 is violated. However, the vanilla SGD is known to converge with the without-replacement strategy [Haochen and Sra, 2019], and some of the analyses of Adam also adopt without-replacement sampling [Zhang et al., 2022, Wang et al., 2022].

Unfortunately, we find that our ADOPT has a counter example, in which ADOPT fails to converge when using without-replacement sampling. For example, when we consider minimizing $f(\theta) = \sum_{i=1}^3 f_i(\theta)$, where $\theta \in [-1, 1]$, $f_1(\theta) = 1.9\theta$ and $f_2(\theta) = f_2(\theta) = -\theta$, it can be easily observed that ADOPT with $\beta_1 = \beta_2 = 0$ fails to converge to the correct solution, i.e., $\theta = 1$.

This non-convergence can be easily avoided by using the with-replacement strategy. Moreover, the difference between with- and without-replacement sampling becomes negligible when $n$ in the finite-sum $\sum_{i=1}^n f_i$ is large enough; hence it does not affect the practical performance very much. In fact, our experiments except for the toy example are performed using without-replacement sampling, but divergent behaviors are not observed. If one applies ADOPT to problems where the difference seems severe (e.g., when training with a small dataset), we recommend to use with-replacement sampling instead of random shuffling for stable training. When one uses PyTorch [Paszke et al., 2019b] for the implementation, for example, with-replacement sampling can be easily applied by specifying `replacemnet=True` for `torch.utils.data.RandomSampler`, and feeding it to the `sampler` argument of `torch.utils.data.DataLoader`.

## C  Another Expression of ADOPT

---

**Algorithm 2** Alternative representation of ADOPT algorithm

---

**Require:** Learning rate $\{\alpha_t\}$, initial parameter $\boldsymbol{\theta}_0$
**Require:** Exponential decay rate $0 \leq \beta_1 < 1, 0 \leq \beta_2 \leq 1$, small constant $\epsilon > 0$

$\quad \boldsymbol{v}_0 \leftarrow \boldsymbol{g}_0 \odot \boldsymbol{g}_0$
$\quad$ **for** $t = 1$ to $T$ **do**
$\quad\quad$ **if** $t = 1$ **then**
$\quad\quad\quad \boldsymbol{m}_t \leftarrow \boldsymbol{g}_t / \max\left\{\sqrt{\boldsymbol{v}_{t-1}}, \epsilon\right\}$
$\quad\quad$ **else**
$\quad\quad\quad \boldsymbol{m}_t \leftarrow \beta_1 \cdot \boldsymbol{m}_{t-1} + (1 - \beta_1)\, \boldsymbol{g}_t / \max\left\{\sqrt{\boldsymbol{v}_{t-1}}, \epsilon\right\}$
$\quad\quad$ **end if**
$\quad\quad \boldsymbol{\theta}_t \leftarrow \boldsymbol{\theta}_{t-1} - \alpha_t \boldsymbol{m}_t$
$\quad\quad \boldsymbol{v}_t \leftarrow \beta_2 \cdot \boldsymbol{v}_{t-1} + (1 - \beta_2)\, \boldsymbol{g}_t \odot \boldsymbol{g}_t$
$\quad$ **end for**
$\quad$ **return** $\{\boldsymbol{\theta}_t\}_{t=1}^{T}$

---

## D   Recommendation of Hyperparameter Settings for ADOPT

We experimentally find that our ADOPT works similarly to Adam when the same hyperparameters are used, but $\epsilon$ should be set to a little larger value (e.g., $1 \times 10^{-6}$) for ADOPT compared to Adam, in which $\epsilon$ is set to $1 \times 10^{-8}$ by default. Our recommendation of the hyperparameter settings for ADOPT is provided in Table 5.

| | |
|---|---|
| $\beta_1$ | 0.9 |
| $\beta_2$ | 0.9999 |
| $\epsilon$ | $1 \times 10^{-6}$ |

Table 5: Recommended hyperparameters for the ADOPT algorithm

## E   Theorems

**Theorem E.1.** *Under Assumptions 2.1, 2.2, 2.3, and 2.5, if the objective $f$ is upper-bounded by $f_{\sup}$, the following holds for the ADOPT algorithm with a learning rate $\alpha_t = \alpha/\sqrt{t}$:*

$$
\min_{t=1,\ldots,T} \left\{ \mathbb{E}\left[ \|\nabla f(\boldsymbol{\theta}_t)\|^{4/3} \right]^{3/2} \right\}
$$
$$
\leq \frac{3\sqrt{\max\{G^2, 1\} + \epsilon^2}}{2\left((T+1)^{3/2} - 1\right)} \left( \frac{f_{\sup} - f_{\inf}}{\alpha}(T+1) + \left( \frac{\sqrt{2}\alpha\beta_1 G^2 L}{\epsilon^2(1-\beta_1)} + \frac{\alpha G^2 L}{2\epsilon^2} \right) T \right)
$$
$$
+ \frac{3\sqrt{\max\{G^2, 1\} + \epsilon^2}}{2\left((T+1)^{3/2} - 1\right)} \left( \frac{2\sqrt{2}\beta_1 G^2}{\epsilon(1-\beta_1)}\left(\sqrt{T+1} - 1\right) + \frac{\alpha\beta_1^2 G^2 L}{\epsilon(1-\beta_1)^2}\frac{T}{T+1} \right)
$$
$$
+ \frac{3\sqrt{\max\{G^2, 1\} + \epsilon^2}}{2\left((T+1)^{3/2} - 1\right)} \left( \frac{2\alpha\beta_1^2 G^2 L}{\epsilon(1-\beta_1)^2} + \frac{\alpha^2 \beta_1 G^2 L}{\sqrt{2}\epsilon^2(1-\beta_1)} \right) \log(T+1). \tag{20}
$$

## F   Proofs

*Proof of Theorems 4.1 and E.1.* We define $\phi_t$ for $t \geq 1$ as follows:

$$
\phi_t = \frac{1}{1-\beta_1}\boldsymbol{\theta}_t - \frac{\beta_1}{1-\beta_1}\boldsymbol{\theta}_{t-1}. \tag{21}
$$

We also define $\phi_0 = \theta_0$. By Assumption 2.3, the following holds for $t \geq 1$:

$$f\left(\phi_t\right) \leq f\left(\phi_{t-1}\right) + \nabla f\left(\phi_{t-1}\right)^\top \left(\phi_t - \phi_{t-1}\right) + \frac{L}{2}\left\|\phi_t - \phi_{t-1}\right\|^2 \tag{22}$$

$$= f\left(\phi_{t-1}\right) + \nabla f\left(\theta_{t-1}\right)^\top \left(\phi_t - \phi_{t-1}\right)$$
$$+ \left(\nabla f\left(\phi_{t-1}\right) - \nabla f\left(\theta_{t-1}\right)\right)^\top \left(\phi_t - \phi_{t-1}\right) + \frac{L}{2}\left\|\phi_t - \phi_{t-1}\right\|^2 \tag{23}$$

$$\leq f\left(\phi_{t-1}\right) + \nabla f\left(\theta_{t-1}\right)^\top \left(\phi_t - \phi_{t-1}\right)$$
$$+ \left\|\nabla f\left(\phi_{t-1}\right) - \nabla f\left(\theta_{t-1}\right)\right\| \left\|\phi_t - \phi_{t-1}\right\| + \frac{L}{2}\left\|\phi_t - \phi_{t-1}\right\|^2 \tag{24}$$

$$\leq f\left(\phi_{t-1}\right) + \nabla f\left(\theta_{t-1}\right)^\top \left(\phi_t - \phi_{t-1}\right)$$
$$+ L\left\|\phi_{t-1} - \theta_{t-1}\right\| \left\|\phi_t - \phi_{t-1}\right\| + \frac{L}{2}\left\|\phi_t - \phi_{t-1}\right\|^2, \tag{25}$$

where the second inequality holds due to the Cauchy-Schwarz inequality, and the last inequality holds due to Assumption 2.3.

By taking the expectation, the following holds:

$$\mathbb{E}\left[f\left(\phi_t\right)\right] \leq \mathbb{E}\left[f\left(\phi_{t-1}\right)\right] + \mathbb{E}\left[\nabla f\left(\theta_{t-1}\right)^\top \left(\phi_t - \phi_{t-1}\right)\right]$$
$$+ L\mathbb{E}\left[\left\|\phi_{t-1} - \theta_{t-1}\right\| \left\|\phi_t - \phi_{t-1}\right\|\right] + \frac{L}{2}\mathbb{E}\left[\left\|\phi_t - \phi_{t-1}\right\|^2\right] \tag{26}$$

$$\leq \mathbb{E}\left[f\left(\phi_{t-1}\right)\right] + \frac{\left(\alpha_{t-1} - \alpha_t\right)\beta_1\left(1 - \beta_1^{t-1}\right)G^2}{\left(1 - \beta_1\right)\epsilon} - \alpha_t \frac{\mathbb{E}\left[\left\|\nabla f\left(\theta_{t-1}\right)\right\|_i^{4/3}\right]^{3/2}}{\sqrt{\left(1 - \beta_2^T\right)G^2 + \epsilon^2}} \tag{27}$$

$$+ \frac{\alpha_{t-1}\left(\alpha_{t-1} - \alpha_t\right)\beta_1^2\left(1 - \beta_1^{t-1}\right)G^2L}{\epsilon^2\left(1 - \beta_1\right)^2} + \frac{\alpha_t\alpha_{t-1}\beta_1\sqrt{1 - \beta_1^{t-1}}G^2L}{\left(1 - \beta_1\right)\epsilon^2}$$

$$+ \frac{\left(\alpha_{t-1} - \alpha_t\right)^2\beta_1^2\left(1 - \beta_1^{t-1}\right)G^2L}{2\left(1 - \beta_1\right)^2\epsilon^2}$$

$$+ \frac{\alpha_t^2G^2L}{2\epsilon^2} + \frac{\alpha_t\left(\alpha_{t-1} - \alpha_t\right)\beta_1\sqrt{1 - \beta_1^{t-1}}G^2L}{2\left(1 - \beta_1\right)\epsilon^2}.$$

When $\alpha_t = \alpha$, the following holds:

$$\mathbb{E}\left[f\left(\phi_t\right)\right] \leq \mathbb{E}\left[f\left(\phi_{t-1}\right)\right] - \alpha\frac{\mathbb{E}\left[\left\|\nabla f\left(\theta_{t-1}\right)\right\|_i^{4/3}\right]^{3/2}}{\sqrt{\left(1 - \beta_2^T\right)G^2 + \epsilon^2}} + \frac{\alpha^2\beta_1\sqrt{1 - \beta_1^{t-1}}G^2L}{\left(1 - \beta_1\right)\epsilon^2} + \frac{\alpha^2G^2L}{2\epsilon^2} \tag{28}$$

$$\leq \mathbb{E}\left[f\left(\phi_{t-1}\right)\right] - \alpha\frac{\mathbb{E}\left[\left\|\nabla f\left(\theta_{t-1}\right)\right\|^{4/3}\right]^{3/2}}{\sqrt{\left(1 - \beta_2^T\right)G^2 + \epsilon^2}} + \frac{\alpha^2\left(1 + \beta_1\right)G^2L}{2\left(1 - \beta_1\right)\epsilon^2}. \tag{29}$$

Telescoping it for $t = 1, \ldots, T$, we have

$$\mathbb{E}\left[f\left(\phi_T\right)\right] \leq f\left(\theta_0\right) - \alpha\frac{\sum_{t=1}^T \mathbb{E}\left[\left\|\nabla f\left(\theta_{t-1}\right)\right\|^{4/3}\right]^{3/2}}{\sqrt{\left(1 - \beta_2^T\right)G^2 + \epsilon^2}} + \frac{\alpha^2\left(1 + \beta_1\right)G^2LT}{2\left(1 - \beta_1\right)\epsilon^2} \tag{30}$$

$$\leq f\left(\theta_0\right) - \alpha\frac{\sum_{t=1}^T \mathbb{E}\left[\left\|\nabla f\left(\theta_{t-1}\right)\right\|^{4/3}\right]^{3/2}}{\sqrt{\left(1 - \beta_2^T\right)G^2 + \epsilon^2}} + \frac{\alpha^2\left(1 + \beta_1\right)G^2LT}{2\left(1 - \beta_1\right)\epsilon^2} \tag{31}$$

By rearranging the terms, we have

$$\min_{t=1,\dots,T}\left\{\mathbb{E}\left[\|\nabla f\left(\boldsymbol{\theta}_{t-1}\right)\|^{4/3}\right]^{3/2}\right\}$$

$$\leq \frac{\sum_{t=1}^{T}\mathbb{E}\left[\|\nabla f\left(\boldsymbol{\theta}_{t-1}\right)\|^{4/3}\right]^{3/2}}{T} \tag{32}$$

$$\leq \sqrt{\left(1-\beta_2^T\right)G^2+\epsilon^2}\left(\frac{f\left(\boldsymbol{\theta}_0\right)-f_{\text{inf}}}{\alpha T}+\frac{\alpha\left(1+\beta_1\right)G^2 L}{2\left(1-\beta_1\right)\epsilon^2}\right) \tag{33}$$

$$\tag{34}$$

When $\alpha_t = \alpha/\sqrt{t}$, the following holds for $t \geq 2$:

$$\alpha_{t-1}-\alpha_t = \alpha\left(\frac{1}{\sqrt{t-1}}-\frac{1}{\sqrt{t}}\right) \tag{35}$$

$$= \frac{\alpha\left(\sqrt{t}-\sqrt{t-1}\right)}{\sqrt{t\left(t-1\right)}} \tag{36}$$

$$= \frac{\alpha}{\sqrt{t\left(t-1\right)}\left(\sqrt{t}+\sqrt{t-1}\right)} \tag{37}$$

$$\leq \frac{\alpha}{2\left(t-1\right)^{3/2}} \tag{38}$$

$$\leq \frac{\sqrt{2}\alpha}{t^{3/2}}. \tag{39}$$

This also holds for $t = 1$ by defining $\alpha_0 = \alpha$. Applying it to Eq. (27), we have

$$\mathbb{E}\left[f\left(\phi_t\right)\right] \leq \mathbb{E}\left[f\left(\phi_{t-1}\right)\right] + \frac{\left(\alpha_{t-1} - \alpha_t\right)\beta_1\left(1 - \beta_1^{t-1}\right)G^2}{\left(1 - \beta_1\right)\epsilon} - \alpha_t \frac{\mathbb{E}\left[\left\|\nabla f\left(\boldsymbol{\theta}_{t-1}\right)\right\|_i^{4/3}\right]^{3/2}}{\sqrt{\left(1 - \beta_2^T\right)G^2 + \epsilon^2}}$$

$$+ \frac{\alpha_{t-1}\left(\alpha_{t-1} - \alpha_t\right)\beta_1^2\left(1 - \beta_1^{t-1}\right)G^2 L}{\epsilon^2\left(1 - \beta_1\right)^2} + \frac{\alpha_t\alpha_{t-1}\beta_1\sqrt{1 - \beta_1^{t-1}}G^2 L}{\left(1 - \beta_1\right)\epsilon^2}$$

$$+ \frac{\left(\alpha_{t-1} - \alpha_t\right)^2\beta_1^2\left(1 - \beta_1^{t-1}\right)G^2 L}{2\left(1 - \beta_1\right)^2\epsilon^2} + \frac{\alpha_t^2 G^2 L}{2\epsilon^2}$$

$$+ \frac{\alpha_t\left(\alpha_{t-1} - \alpha_t\right)\beta_1\sqrt{1 - \beta_1^{t-1}}G^2 L}{2\left(1 - \beta_1\right)\epsilon^2} \tag{40}$$

$$\leq \mathbb{E}\left[f\left(\phi_{t-1}\right)\right] + \frac{\sqrt{2}\alpha\beta_1\left(1 - \beta_1^{t-1}\right)G^2}{t^{3/2}\left(1 - \beta_1\right)\epsilon} - \frac{\alpha}{\sqrt{t}}\frac{\mathbb{E}\left[\left\|\nabla f\left(\boldsymbol{\theta}_{t-1}\right)\right\|_i^{4/3}\right]^{3/2}}{\sqrt{\left(1 - \beta_2^T\right)G^2 + \epsilon^2}}$$

$$+ \frac{2\alpha^2\beta_1^2 G^2 L}{\epsilon^2\left(1 - \beta_1\right)^2 t^2} + \frac{\sqrt{2}\alpha^2\beta_1 G^2 L}{\left(1 - \beta_1\right)\epsilon^2 t} + \frac{\alpha^2\beta_1^2 G^2 L}{\left(1 - \beta_1\right)^2\epsilon^2 t^3} + \frac{\alpha^2 G^2 L}{2\epsilon^2 t}$$

$$+ \frac{\alpha^2\beta_1 G^2 L}{\sqrt{2}\left(1 - \beta_1\right)\epsilon^2 t^2} \tag{41}$$

$$= \mathbb{E}\left[f\left(\phi_{t-1}\right)\right] - \frac{\alpha}{\sqrt{t}}\frac{\mathbb{E}\left[\left\|\nabla f\left(\boldsymbol{\theta}_{t-1}\right)\right\|_i^{4/3}\right]^{3/2}}{\sqrt{\left(1 - \beta_2^T\right)G^2 + \epsilon^2}}$$

$$+ \frac{\alpha^2\left(1 + \left(2\sqrt{2} - 1\right)\beta_1\right)G^2 L}{2\left(1 - \beta_1\right)\epsilon^2}\cdot t^{-1} + \frac{\sqrt{2}\alpha\beta_1 G^2}{\left(1 - \beta_1\right)\epsilon}\cdot t^{-\frac{3}{2}}$$

$$+ \frac{\alpha^2\beta_1\left(1 + \left(2\sqrt{2} - 1\right)\beta_1\right)G^2 L}{\sqrt{2}\left(1 - \beta_1\right)^2\epsilon^2}\cdot t^{-2} + \frac{\alpha^2\beta_1^2 G^2 L}{\left(1 - \beta_1\right)^2\epsilon^2}\cdot t^{-3}. \tag{42}$$

Multiplying $t$ to the both sides and rearranging the terms, we have

$$\frac{\sqrt{t}\,\mathbb{E}\left[\left\|\nabla f\left(\boldsymbol{\theta}_{t-1}\right)\right\|^{4/3}\right]^{3/2}}{\sqrt{\left(1 - \beta_2^T\right)G^2 + \epsilon^2}}$$

$$\leq \frac{\mathbb{E}\left[f\left(\phi_{t-1}\right) - f\left(\phi_t\right)\right]}{\alpha}\cdot t + \frac{\alpha\left(1 + \left(2\sqrt{2} - 1\right)\beta_1\right)G^2 L}{2\left(1 - \beta_1\right)\epsilon^2} + \frac{\sqrt{2}\beta_1 G^2}{\left(1 - \beta_1\right)\epsilon}\cdot t^{-\frac{1}{2}} \tag{43}$$

$$+ \frac{\alpha\beta_1\left(1 + \left(2\sqrt{2} - 1\right)\beta_1\right)G^2 L}{\sqrt{2}\left(1 - \beta_1\right)^2\epsilon^2}t^{-1} + \frac{\alpha\beta_1^2 G^2 L}{\left(1 - \beta_1\right)^2\epsilon^2}t^{-2} \tag{44}$$

$$\sum_{t=1}^{T} \frac{\sqrt{t}\, \mathbb{E}\left[\|\nabla f\left(\boldsymbol{\theta}_{t-1}\right)\|^{4/3}\right]^{3/2}}{\sqrt{\left(1-\beta_2^T\right) G^2 + \epsilon^2}}$$

$$\leq \frac{f\left(\boldsymbol{\phi}_0\right) - T f\left(\boldsymbol{\phi}_T\right) + \sum_{t=1}^{T-1} f\left(\boldsymbol{\phi}_t\right)}{\alpha} + \frac{\alpha\left(1+\left(2\sqrt{2}-1\right)\beta_1\right) G^2 L T}{2\left(1-\beta_1\right)\epsilon^2}$$

$$+ \frac{\sqrt{2}\beta_1 G^2}{\left(1-\beta_1\right)\epsilon} \sum_{t=1}^{T} t^{-\frac{1}{2}} + \frac{\alpha\beta_1\left(1+\left(2\sqrt{2}-1\right)\beta_1\right) G^2 L}{\sqrt{2}\left(1-\beta_1\right)^2 \epsilon^2} \sum_{t=1}^{T} t^{-1} + \frac{\alpha\beta_1^2 G^2 L}{\left(1-\beta_1\right)^2 \epsilon^2} \sum_{t=1}^{T} t^{-2} \quad (45)$$

$$\leq \frac{f_{\sup} - f_{\inf}}{\alpha} T + \frac{\alpha\left(1+\left(2\sqrt{2}-1\right)\beta_1\right) G^2 L T}{2\left(1-\beta_1\right)\epsilon^2}$$

$$+ \frac{\sqrt{2}\beta_1 G^2}{\left(1-\beta_1\right)\epsilon}\left(1+\int_1^T t^{-\frac{1}{2}} dt\right) + \frac{\alpha\beta_1\left(1+\left(2\sqrt{2}-1\right)\beta_1\right) G^2 L}{\sqrt{2}\left(1-\beta_1\right)^2 \epsilon^2}\left(1+\int_1^T t^{-1} dt\right)$$

$$+ \frac{\alpha\beta_1^2 G^2 L}{\left(1-\beta_1\right)^2 \epsilon^2}\left(1+\int_1^T t^{-2} dt\right) \quad (46)$$

$$\leq \frac{f_{\sup} - f_{\inf}}{\alpha} T + \frac{\alpha\left(1+\left(2\sqrt{2}-1\right)\beta_1\right) G^2 L T}{2\left(1-\beta_1\right)\epsilon^2} + \frac{\sqrt{2}\beta_1 G^2}{\left(1-\beta_1\right)\epsilon}\left(2\sqrt{T}-1\right)$$

$$+ \frac{\alpha\beta_1\left(1+\left(2\sqrt{2}-1\right)\beta_1\right) G^2 L}{\sqrt{2}\left(1-\beta_1\right)^2 \epsilon^2}\left(1+\log T\right) + \frac{\alpha\beta_1^2 G^2 L}{\left(1-\beta_1\right)^2 \epsilon^2}\left(2-\frac{1}{T}\right) \quad (47)$$

Therefore, the following bound is derived.

$$\min_{t=1,\ldots,T}\left\{\mathbb{E}\left[\|\nabla f\left(\boldsymbol{\theta}_{t-1}\right)\|^{4/3}\right]^{3/2}\right\}$$

$$\leq \frac{\sum_{t=1}^{T} \sqrt{t}\, \mathbb{E}\left[\|\nabla f\left(\boldsymbol{\theta}_{t-1}\right)\|^{4/3}\right]^{3/2}}{\sum_{t=1}^{T} \sqrt{t}} \quad (48)$$

$$\leq \frac{\sum_{t=1}^{T} \sqrt{t}\, \mathbb{E}\left[\|\nabla f\left(\boldsymbol{\theta}_{t-1}\right)\|^{4/3}\right]^{3/2}}{\int_0^T \sqrt{t}\, dt} \quad (49)$$

$$\leq \frac{3 C_T\left(f_{\sup} - f_{\inf}\right)}{2\alpha} \frac{1}{\sqrt{T}} + \frac{3\alpha\left(1+\left(2\sqrt{2}-1\right)\beta_1\right) C_T G^2 L}{4\left(1-\beta_1\right)\epsilon^2 \sqrt{T}} + \frac{3\beta_1 C_T G^2}{\sqrt{2}\left(1-\beta_1\right)\epsilon}\left(\frac{2}{T}-\frac{1}{T^{3/2}}\right)$$

$$+ \frac{3\alpha\beta_1\left(1+\left(2\sqrt{2}-1\right)\beta_1\right) C_T G^2 L}{2\sqrt{2}\left(1-\beta\right)^2 \epsilon^2}\left(\frac{1}{T^{3/2}}+\frac{\log T}{T^{3/2}}\right) + \frac{3\alpha\beta_1^2 C_T G^2 L}{2\left(1-\beta_1\right)^2 \epsilon^2}\left(\frac{2}{T^{3/2}}-\frac{1}{T^{5/2}}\right), \quad (50)$$

where $C_T = \sqrt{\left(1-\beta_2^T\right) G^2 + \epsilon^2}$.

$$\square$$

# G   Lemmas

**Lemma G.1.** *For all $\boldsymbol{\theta} \in \mathbb{R}^D$ and $t \geq 1$, the following holds*

$$\|\nabla f\left(\boldsymbol{\theta}_{t-1}\right)\| \leq G. \quad (51)$$

*Proof.*

$$\|\nabla f\left(\boldsymbol{\theta}_{t-1}\right)\| = \sqrt{\|\mathbb{E}\left[\boldsymbol{g}_t\right]\|^2} \tag{52}$$

$$\leq \sqrt{\mathbb{E}\left[\|\boldsymbol{g}_t\|^2\right]} \tag{53}$$

$$\leq G. \tag{54}$$

The first inequality holds because $\mathbb{E}[(\boldsymbol{g}_t)_i]^2 \leq \mathbb{E}[(\boldsymbol{g}_t)_i^2]$, and the second inequality holds due to Assumption 2.5. □

**Lemma G.2.** *For all $\boldsymbol{\theta} \in \mathbb{R}^D$ and $t \geq 1$, the following holds*

$$\mathbb{E}\left[\|\boldsymbol{g}_t\|\right] \leq G \tag{55}$$

*Proof.*

$$\mathbb{E}\left[\|\boldsymbol{g}_t\|\right] \leq \mathbb{E}\left[\|\boldsymbol{g}_t\|^2\right]^{1/2} \tag{56}$$

$$\leq G, \tag{57}$$

where the first inequality holds due to the Hölder's inequality and the second one holds due to Assumption 2.5. □

**Lemma G.3.** *For the RMSprop algorithm, the following holds for $t \geq 1$:*

$$\mathbb{E}\left[\sum_{i=1}^{D}\left(\boldsymbol{v}_t\right)_i\right] \leq \left(1 - \beta_2^t\right) G^2 \tag{58}$$

*Proof.*

$$\mathbb{E}\left[\sum_{i=1}^{D}\left(\boldsymbol{v}_t\right)_i\right] = \mathbb{E}\left[(1-\beta_2)\sum_{i=1}^{D}\sum_{k=1}^{t}\beta_2^{t-k}\left(\boldsymbol{g}_k\right)_i^2\right] \tag{59}$$

$$\leq (1-\beta_2) G^2 \sum_{k=1}^{t}\beta_2^{t-k} \tag{60}$$

$$= \left(1 - \beta_2^t\right) G^2. \tag{61}$$

□

**Lemma G.4.** *For the RMSprop algorithm, the following holds:*

$$\mathbb{E}\left[\nabla f\left(\boldsymbol{\theta}_{t-1}\right)^{\top}\left(\frac{\boldsymbol{g}_t}{\sqrt{\boldsymbol{v}_t + \epsilon^2}}\right)\right]$$

$$\geq \frac{1}{2}\mathbb{E}\left[\nabla f\left(\boldsymbol{\theta}_{t-1}\right)^{\top}\left(\frac{\boldsymbol{g}_t}{\sqrt{\tilde{\boldsymbol{v}}_t + \epsilon^2}}\right)\right] - 2G\sqrt{1-\beta_2}\mathbb{E}\left[\left\|\frac{\boldsymbol{g}_t}{\sqrt{\boldsymbol{v}_t + \epsilon^2}}\right\|^2\right] \tag{62}$$

*Proof.*

$$\mathbb{E}\left[\nabla f\left(\boldsymbol{\theta}_{t-1}\right)^{\top}\left(\frac{\boldsymbol{g}_t}{\sqrt{\boldsymbol{v}_t + \epsilon^2}}\right)\right] = \sum_{i=1}^{D}\mathbb{E}\left[\frac{\left(\nabla f\left(\boldsymbol{\theta}_{t-1}\right)\right)_i\left(\boldsymbol{g}_t\right)_i}{\sqrt{\left(\boldsymbol{v}_t\right)_i + \epsilon^2}}\right] \tag{63}$$

We define $\tilde{\boldsymbol{v}}_t$ as follows:

$$\tilde{\boldsymbol{v}}_t = \beta_2\boldsymbol{v}_{t-1} + (1-\beta_2)\mathbb{E}\left[\boldsymbol{g}_t \odot \boldsymbol{g}_t\right] \tag{64}$$

Using this, the following holds:

$$\mathbb{E}\left[\frac{(\nabla f\left(\boldsymbol{\theta}_{t-1}\right))_i\left(\boldsymbol{g}_t\right)_i}{\sqrt{\left(\boldsymbol{v}_t\right)_i + \epsilon^2}}\right]$$

$$= \mathbb{E}\left[\frac{(\nabla f\left(\boldsymbol{\theta}_{t-1}\right))_i\left(\boldsymbol{g}_t\right)_i}{\sqrt{\left(\tilde{\boldsymbol{v}}_t\right)_i + \epsilon^2}}\right] + \mathbb{E}\left[(\nabla f\left(\boldsymbol{\theta}_{t-1}\right))_i\left(\boldsymbol{g}_t\right)_i\left(\frac{1}{\sqrt{\left(\boldsymbol{v}_t\right)_i + \epsilon^2}} - \frac{1}{\sqrt{\left(\tilde{\boldsymbol{v}}_t\right)_i + \epsilon^2}}\right)\right] \quad (65)$$

$$= \mathbb{E}\left[\frac{(\nabla f\left(\boldsymbol{\theta}_{t-1}\right))_i^2}{\sqrt{\left(\tilde{\boldsymbol{v}}_t\right)_i + \epsilon^2}}\right] + \mathbb{E}\left[(\nabla f\left(\boldsymbol{\theta}_{t-1}\right))_i\left(\boldsymbol{g}_t\right)_i\left(\frac{1}{\sqrt{\left(\boldsymbol{v}_t\right)_i + \epsilon^2}} - \frac{1}{\sqrt{\left(\tilde{\boldsymbol{v}}_t\right)_i + \epsilon^2}}\right)\right] \quad (66)$$

$$\geq \mathbb{E}\left[\frac{(\nabla f\left(\boldsymbol{\theta}_{t-1}\right))_i^2}{\sqrt{\left(\tilde{\boldsymbol{v}}_t\right)_i + \epsilon^2}}\right] - \mathbb{E}\left[\left|(\nabla f\left(\boldsymbol{\theta}_{t-1}\right))_i\left(\boldsymbol{g}_t\right)_i\left(\frac{1}{\sqrt{\left(\boldsymbol{v}_t\right)_i + \epsilon^2}} - \frac{1}{\sqrt{\left(\tilde{\boldsymbol{v}}_t\right)_i + \epsilon^2}}\right)\right|\right], \quad (67)$$

where the last inequality holds due to $A \geq -|A|$. For the second term, the following holds:

$$\left|(\nabla f\left(\boldsymbol{\theta}_{t-1}\right))_i\left(\boldsymbol{g}_t\right)_i\left(\frac{1}{\sqrt{\left(\boldsymbol{v}_t\right)_i + \epsilon^2}} - \frac{1}{\sqrt{\left(\tilde{\boldsymbol{v}}_t\right)_i + \epsilon^2}}\right)\right|$$

$$= (1 - \beta_2)\left|(\nabla f\left(\boldsymbol{\theta}_{t-1}\right))_i\left(\boldsymbol{g}_t\right)_i\frac{\mathbb{E}\left[\left(\boldsymbol{g}_t\right)_i^2\right] - \left(\boldsymbol{g}_t\right)_i^2}{\sqrt{\left(\boldsymbol{v}_t\right)_i + \epsilon^2}\sqrt{\left(\tilde{\boldsymbol{v}}_t\right)_i + \epsilon^2}\left(\sqrt{\left(\boldsymbol{v}_t\right)_i + \epsilon^2} + \sqrt{\left(\tilde{\boldsymbol{v}}_t\right)_i + \epsilon^2}\right)}\right| \quad (68)$$

$$\leq (1 - \beta_2)\left(\frac{|(\nabla f\left(\boldsymbol{\theta}_{t-1}\right))_i\left(\boldsymbol{g}_t\right)_i|\,\mathbb{E}\left[\left(\boldsymbol{g}_t\right)_i^2\right]}{\sqrt{\left(\boldsymbol{v}_t\right)_i + \epsilon^2}\left(\left(\tilde{\boldsymbol{v}}_t\right)_i + \epsilon^2\right)} + \frac{|(\nabla f\left(\boldsymbol{\theta}_{t-1}\right))_i\left(\boldsymbol{g}_t\right)_i|\left(\boldsymbol{g}_t\right)_i^2}{\left(\left(\boldsymbol{v}_t\right)_i + \epsilon^2\right)\sqrt{\left(\tilde{\boldsymbol{v}}_t\right)_i + \epsilon^2}}\right), \quad (69)$$

where the last inequality holds due to the triangle inequality. For the first term, the following holds:

$$\mathbb{E}\left[\frac{|(\nabla f\left(\boldsymbol{\theta}_{t-1}\right))_i\left(\boldsymbol{g}_t\right)_i|\,\mathbb{E}\left[\left(\boldsymbol{g}_t\right)_i^2\right]}{\sqrt{\left(\boldsymbol{v}_t\right)_i + \epsilon^2}\left(\left(\tilde{\boldsymbol{v}}_t\right)_i + \epsilon^2\right)}\right]$$

$$\leq \frac{1}{(1 - \beta_2)}\mathbb{E}\left[\frac{(\nabla f\left(\boldsymbol{\theta}_{t-1}\right))_i^2}{4\sqrt{\left(\tilde{\boldsymbol{v}}_t\right)_i + \epsilon^2}}\right] + (1 - \beta_2)\mathbb{E}\left[\frac{\left(\boldsymbol{g}_t\right)_i^2\,\mathbb{E}\left[\left(\boldsymbol{g}_t\right)_i^2\right]^2}{\left(\left(\boldsymbol{v}_t\right)_i + \epsilon^2\right)\left(\left(\tilde{\boldsymbol{v}}_t\right)_i + \epsilon^2\right)^{3/2}}\right] \quad (70)$$

$$\leq \frac{1}{(1 - \beta_2)}\mathbb{E}\left[\frac{(\nabla f\left(\boldsymbol{\theta}_{t-1}\right))_i^2}{4\sqrt{\left(\tilde{\boldsymbol{v}}_t\right)_i + \epsilon^2}}\right] + \mathbb{E}\left[\frac{\left(\boldsymbol{g}_t\right)_i^2\sqrt{\mathbb{E}\left[\left(\boldsymbol{g}_t\right)_i^2\right]}}{\sqrt{1 - \beta_2}\left(\left(\boldsymbol{v}_t\right)_i + \epsilon^2\right)}\right] \quad (71)$$

$$\leq \frac{1}{(1 - \beta_2)}\mathbb{E}\left[\frac{(\nabla f\left(\boldsymbol{\theta}_{t-1}\right))_i^2}{4\sqrt{\left(\tilde{\boldsymbol{v}}_t\right)_i + \epsilon^2}}\right] + \frac{G}{\sqrt{1 - \beta_2}}\mathbb{E}\left[\frac{\left(\boldsymbol{g}_t\right)_i^2}{\left(\boldsymbol{v}_t\right)_i + \epsilon^2}\right] \quad (72)$$

The first inequality is derived using the following fact:

$$\forall \lambda > 0, x, y \in \mathbb{R}, xy \leq \frac{\lambda}{2}x^2 + \frac{y^2}{2\lambda}. \quad (73)$$

For the second term of Eq. (69), the following holds:

$$\mathbb{E}\left[\frac{|(\nabla f(\boldsymbol{\theta}_{t-1}))_i(\boldsymbol{g}_t)_i|(\boldsymbol{g}_t)_i^2}{((\boldsymbol{v}_t)_i + \epsilon^2)\sqrt{(\tilde{\boldsymbol{v}}_t)_i + \epsilon^2}}\right] \tag{74}$$

$$\leq \frac{1}{(1-\beta_2)}\mathbb{E}\left[\frac{(\nabla f(\boldsymbol{\theta}_{t-1}))_i^2}{4\sqrt{(\tilde{\boldsymbol{v}}_t)_i + \epsilon^2}}\frac{(\boldsymbol{g}_t)_i^2}{\mathbb{E}\left[(\boldsymbol{g}_t)_i^2\right]}\right] + (1-\beta_2)\mathbb{E}\left[\frac{\mathbb{E}\left[(\boldsymbol{g}_t)_i^2\right]}{\sqrt{(\tilde{\boldsymbol{v}}_t)_i + \epsilon^2}}\frac{(\boldsymbol{g}_t)_i^4}{((\tilde{\boldsymbol{v}}_t)_i + \epsilon^2)^2}\right] \tag{75}$$

$$\leq \frac{1}{(1-\beta_2)}\mathbb{E}\left[\frac{(\nabla f(\boldsymbol{\theta}_{t-1}))_i^2}{4\sqrt{(\tilde{\boldsymbol{v}}_t)_i + \epsilon^2}}\right] + \mathbb{E}\left[\frac{\sqrt{\mathbb{E}\left[(\boldsymbol{g}_t)_i^2\right]}(\boldsymbol{g}_t)_i^2}{\sqrt{1-\beta_2}((\tilde{\boldsymbol{v}}_t)_i + \epsilon^2)}\right] \tag{76}$$

$$\leq \frac{1}{(1-\beta_2)}\mathbb{E}\left[\frac{(\nabla f(\boldsymbol{\theta}_{t-1}))_i^2}{4\sqrt{(\tilde{\boldsymbol{v}}_t)_i + \epsilon^2}}\right] + \frac{G}{\sqrt{1-\beta_2}}\mathbb{E}\left[\frac{(\boldsymbol{g}_t)_i^2}{(\boldsymbol{v}_t)_i + \epsilon^2}\right] \tag{77}$$

The first inequality is derived using Eq. (73).

Putting these inequalities together, the following is derived:

$$\mathbb{E}\left[\nabla f(\boldsymbol{\theta}_{t-1})^\top\left(\frac{\boldsymbol{g}_t}{\sqrt{\boldsymbol{v}_t + \epsilon^2}}\right)\right]$$

$$\geq \sum_{i=1}^{D}\mathbb{E}\left[\frac{(\nabla f(\boldsymbol{\theta}_{t-1}))_i^2}{2\sqrt{(\tilde{\boldsymbol{v}}_t)_i + \epsilon^2}}\right] - 2G\sqrt{1-\beta_2}\mathbb{E}\left[\frac{(\boldsymbol{g}_t)_i^2}{(\boldsymbol{v}_t)_i + \epsilon^2}\right] \tag{78}$$

$$\geq \frac{1}{2}\mathbb{E}\left[\nabla f(\boldsymbol{\theta}_{t-1})^\top\left(\frac{\boldsymbol{g}_t}{\sqrt{\tilde{\boldsymbol{v}}_t + \epsilon^2}}\right)\right] - 2G\sqrt{1-\beta_2}\mathbb{E}\left[\left\|\frac{\boldsymbol{g}_t}{\sqrt{\boldsymbol{v}_t + \epsilon^2}}\right\|^2\right]. \tag{79}$$

$\square$

**Lemma G.5.** *For the RMSprop algorithm, the following holds:*

$$\sum_{t=1}^{T}\mathbb{E}\left[\left\|\frac{\boldsymbol{g}_t}{\sqrt{\boldsymbol{v}_t + \epsilon^2}}\right\|^2\right] \leq D\left(\log\left(1 + \frac{(1-\beta_2^T)G^2}{\epsilon^2}\right) - T\log\beta_2\right) \tag{80}$$

*Proof.*

$$\left\|\frac{\boldsymbol{g}_t}{\sqrt{\boldsymbol{v}_t + \epsilon^2}}\right\|^2 = \sum_{i=1}^{D}\frac{(\boldsymbol{g}_t)_i^2}{(\boldsymbol{v}_t)_i + \epsilon^2} \tag{81}$$

$$\frac{(\boldsymbol{g}_t)_i^2}{(\boldsymbol{v}_t)_i + \epsilon^2} = \frac{1}{1-\beta_2}\frac{(1-\beta_2)(\boldsymbol{g}_t)_i^2}{(\boldsymbol{v}_t)_i + \epsilon^2} \tag{82}$$

$$\leq -\frac{1}{1-\beta_2}\log\left(1 - \frac{(1-\beta_2)(\boldsymbol{g}_t)_i^2}{(\boldsymbol{v}_t)_i + \epsilon^2}\right) \tag{83}$$

$$= \frac{1}{1-\beta_2}\log\left(\frac{(\boldsymbol{v}_t)_i + \epsilon^2}{\beta_2(\boldsymbol{v}_{t-1})_i + \epsilon^2}\right) \tag{84}$$

$$= \frac{1}{1-\beta_2}\left(\log\left(\frac{(\boldsymbol{v}_t)_i + \epsilon^2}{(\boldsymbol{v}_{t-1})_i + \epsilon^2}\right) + \log\left(\frac{(\boldsymbol{v}_{t-1})_i + \epsilon^2}{\beta_2(\boldsymbol{v}_{t-1})_i + \epsilon^2}\right)\right) \tag{85}$$

$$\leq \frac{1}{1-\beta_2}\left(\log\left(\frac{(\boldsymbol{v}_t)_i + \epsilon^2}{(\boldsymbol{v}_{t-1})_i + \epsilon^2}\right) - \log\beta_2\right) \tag{86}$$

$$\sum_{t=1}^{T} \frac{(\boldsymbol{g}_t)_i^2}{(\boldsymbol{v}_t)_i + \epsilon^2} \leq \frac{1}{1 - \beta_2} \left( \log \left( \frac{(\boldsymbol{v}_T)_i + \epsilon^2}{\epsilon^2} \right) - T \log \beta_2 \right) \tag{87}$$

$$\leq \frac{1}{1 - \beta_2} \left( \log \left( 1 + \frac{\left(1 - \beta_2^T\right) G^2}{\epsilon^2} \right) - T \log \beta_2 \right) \tag{88}$$

$$\sum_{t=1}^{T} \mathbb{E} \left[ \left\| \frac{\boldsymbol{g}_t}{\sqrt{\boldsymbol{v}_t + \epsilon^2}} \right\|^2 \right] \leq \sum_{i=1}^{D} \mathbb{E} \left[ \sum_{t=1}^{T} \frac{(\boldsymbol{g}_t)_i^2}{(\boldsymbol{v}_t)_i + \epsilon^2} \right] \tag{89}$$

$$\leq \frac{1}{1 - \beta_2} \sum_{i=1}^{D} \mathbb{E} \left[ \log \left( 1 + \frac{(\boldsymbol{v}_T)_i}{\epsilon^2} \right) \right] - \frac{DT \log \beta_2}{1 - \beta_2} \tag{90}$$

$$\leq \sum_{i=1}^{D} \log \left( 1 + \frac{\mathbb{E}\left[ (\boldsymbol{v}_T)_i \right]}{\epsilon^2} \right) - \frac{DT \log \beta_2}{1 - \beta_2} \tag{91}$$

$$\leq \frac{D}{1 - \beta_2} \left( \log \left( 1 + \frac{\left(1 - \beta_2^T\right) G^2}{\epsilon^2} \right) - T \log \beta_2 \right) \tag{92}$$

$$\square$$

**Lemma G.6.** *For the RMSprop algorithm, the following holds:*

$$\mathbb{E} \left[ \nabla f \left( \boldsymbol{\theta}_{t-1} \right)^\top \left( \frac{\boldsymbol{g}_t}{\sqrt{\beta_2 \tilde{\boldsymbol{v}}_t + \epsilon^2}} \right) \right] \geq \frac{\mathbb{E} \left[ \| \nabla f \left( \boldsymbol{\theta}_{t-1} \right) \|^{4/3} \right]^{3/2}}{\sqrt{\left(1 - \beta_2^t\right) G^2 + \epsilon^2}} \tag{93}$$

*Proof.*

$$\mathbb{E} \left[ \nabla f \left( \boldsymbol{\theta}_{t-1} \right)^\top \left( \frac{\boldsymbol{g}_t}{\sqrt{\tilde{\boldsymbol{v}}_t + \epsilon^2}} \right) \right]$$

$$= \sum_{i=1}^{D} \mathbb{E} \left[ \frac{\left( \nabla f \left( \boldsymbol{\theta}_{t-1} \right) \right)_i \cdot (\boldsymbol{g}_t)_i}{\sqrt{(\tilde{\boldsymbol{v}}_t)_i + \epsilon^2}} \right]$$

$$= \sum_{i=1}^{D} \mathbb{E} \left[ \frac{\left( \nabla f \left( \boldsymbol{\theta}_{t-1} \right) \right)_i^2}{\sqrt{\beta_2 (\boldsymbol{v}_{t-1})_i + \epsilon^2}} \right]$$

$$\geq \mathbb{E} \left[ \frac{\| \nabla f \left( \boldsymbol{\theta}_{t-1} \right) \|^2}{\sqrt{\sum_{i=1}^{D} (\tilde{\boldsymbol{v}}_t)_i + \epsilon^2}} \right]$$

$$\geq \frac{\mathbb{E} \left[ \| \nabla f \left( \boldsymbol{\theta}_{t-1} \right) \|^{4/3} \right]^{3/2}}{\sqrt{\mathbb{E} \left[ \sum_{i=1}^{D} (\tilde{\boldsymbol{v}}_t)_i \right] + \epsilon^2}}$$

$$\geq \frac{\mathbb{E} \left[ \| \nabla f \left( \boldsymbol{\theta}_{t-1} \right) \|^{4/3} \right]^{3/2}}{\sqrt{\left(1 - \beta_2^t\right) G^2 + \epsilon^2}}. \tag{94}$$

The second equality holds due to Assumption 2.2. The first inequality holds because $(\tilde{\boldsymbol{v}}_t)_i \geq 0$ for all $i = 1, \ldots, D$. The second inequality holds due to the Hölder's inequality. The last inequality holds due to Lemma G.3. $\square$

**Lemma G.7.** *For the ADOPT algorithm, the following holds for $t \geq 1$:*

$$\boldsymbol{\phi}_t - \boldsymbol{\phi}_{t-1} = \frac{(\alpha_{t-1} - \alpha_t) \beta_1}{1 - \beta_1} \boldsymbol{m}_{t-1} - \alpha_t \frac{\boldsymbol{g}_t}{\max \left\{ \sqrt{\boldsymbol{v}_{t-1}}, \epsilon \right\}}, \tag{95}$$

*where we define $\alpha_0 = \alpha$.*

*Proof.* For $t = 1$, the following holds by definition:

$$\phi_1 - \phi_0 = \frac{1}{1 - \beta_1}\boldsymbol{\theta}_1 - \left(\frac{\beta_1}{1 - \beta_1} + 1\right)\boldsymbol{\theta}_0 \tag{96}$$

$$= \frac{1}{1 - \beta_1}\left(\boldsymbol{\theta}_1 - \boldsymbol{\theta}_0\right) \tag{97}$$

$$= -\frac{\alpha_1 \boldsymbol{g}_1}{\max\left\{\sqrt{\boldsymbol{v}_0}, \epsilon\right\}}. \tag{98}$$

For $t \geq 2$, the following holds:

$$\phi_t - \phi_{t-1} = \frac{1}{1 - \beta_1}\left(\boldsymbol{\theta}_t - \boldsymbol{\theta}_{t-1}\right) - \frac{\beta_1}{1 - \beta_1}\left(\boldsymbol{\theta}_{t-1} - \boldsymbol{\theta}_{t-2}\right) \tag{99}$$

$$= \frac{1}{1 - \beta_1}\left(\alpha_{t-1}\beta_1 \boldsymbol{m}_{t-1} - \alpha_t \boldsymbol{m}_t\right) \tag{100}$$

$$= \frac{1}{1 - \beta_1}\left(\alpha_{t-1}\beta_1 \boldsymbol{m}_{t-1} - \alpha_t\left(\beta_1 \boldsymbol{m}_{t-1} + (1 - \beta_1)\frac{\boldsymbol{g}_t}{\max\left\{\sqrt{\boldsymbol{v}_{t-1}}, \epsilon\right\}}\right)\right) \tag{101}$$

$$= \frac{1}{1 - \beta_1}\left((\alpha_{t-1} - \alpha_t)\beta_1 \boldsymbol{m}_{t-1} - \alpha_t(1 - \beta_1)\frac{\boldsymbol{g}_t}{\max\left\{\sqrt{\boldsymbol{v}_{t-1}}, \epsilon\right\}}\right) \tag{102}$$

$$= \frac{(\alpha_{t-1} - \alpha_t)\beta_1}{1 - \beta_1}\boldsymbol{m}_{t-1} - \alpha_t\frac{\boldsymbol{g}_t}{\max\left\{\sqrt{\boldsymbol{v}_{t-1}}, \epsilon\right\}} \tag{103}$$

$\square$

**Lemma G.8.** *For the ADOPT algorithm, the following holds for $t \geq 1$:*

$$\phi_{t-1} - \boldsymbol{\theta}_{t-1} = -\frac{\alpha_{t-1}\beta_1}{1 - \beta_1}\boldsymbol{m}_{t-1}. \tag{104}$$

*Proof.* For $t = 1$, Eq. (104) holds obviously because $\phi_0 = \boldsymbol{\theta}_0$ and $\boldsymbol{m}_0 = \boldsymbol{0}$. For $t \geq 2$, the following holds:

$$\phi_{t-1} - \boldsymbol{\theta}_{t-1} = \left(\frac{1}{1 - \beta_1} - 1\right)\boldsymbol{\theta}_{t-1} - \frac{\beta_1}{1 - \beta_1}\boldsymbol{\theta}_{t-2} \tag{105}$$

$$= \frac{\beta_1}{1 - \beta_1}\left(\boldsymbol{\theta}_{t-1} - \boldsymbol{\theta}_{t-2}\right) \tag{106}$$

$$= -\frac{\alpha_{t-1}\beta_1}{1 - \beta_1}\boldsymbol{m}_{t-1}. \tag{107}$$

$\square$

**Lemma G.9.** *For the ADOPT algorithm, the following holds for $t \geq 1$:*

$$\mathbb{E}\left[\nabla f\left(\boldsymbol{\theta}_{t-1}\right)^\top \left(\phi_t - \phi_{t-1}\right)\right]$$

$$\leq \frac{(\alpha_{t-1} - \alpha_t)\beta_1\left(1 - \beta_1^{t-1}\right)G^2}{(1 - \beta_1)\sqrt{\beta_2^{t-2} + \epsilon^2}} - \alpha_t \frac{\mathbb{E}\left[\|\nabla f\left(\boldsymbol{\theta}_{t-1}\right)\|_i^{4/3}\right]^{3/2}}{\sqrt{(1 - \beta_2^t)G^2 + \epsilon^2}}. \tag{108}$$

*Proof.*

$$\nabla f\left(\boldsymbol{\theta}_{t-1}\right)^\top \left(\phi_t - \phi_{t-1}\right)$$

$$= \frac{(\alpha_{t-1} - \alpha_t)\beta_1}{1 - \beta_1}\nabla f\left(\boldsymbol{\theta}_{t-1}\right))^\top \boldsymbol{m}_{t-1} - \alpha_t \nabla f\left(\boldsymbol{\theta}_{t-1}\right)^\top \frac{\boldsymbol{g}_t}{\max\left\{\sqrt{\boldsymbol{v}_{t-1}}, \epsilon\right\}} \tag{109}$$

$$\leq \frac{(\alpha_{t-1} - \alpha_t)\beta_1}{1 - \beta_1}\|\nabla f\left(\boldsymbol{\theta}_{t-1}\right))\|\|\boldsymbol{m}_{t-1}\| - \alpha_t \nabla f\left(\boldsymbol{\theta}_{t-1}\right)^\top \frac{\boldsymbol{g}_t}{\max\left\{\sqrt{\boldsymbol{v}_{t-1}}, \epsilon\right\}} \tag{110}$$

$$\leq \frac{(\alpha_{t-1} - \alpha_t)\beta_1 G}{1 - \beta_1}\|\boldsymbol{m}_{t-1}\| - \alpha_t \nabla f\left(\boldsymbol{\theta}_{t-1}\right)^\top \frac{\boldsymbol{g}_t}{\max\left\{\sqrt{\boldsymbol{v}_{t-1}}, \epsilon\right\}}. \tag{111}$$

By taking the expectation for both sides, the following holds:

$$\mathbb{E}\left[\nabla f\left(\boldsymbol{\theta}_{t-1}\right)^{\top}\cdot\left(\boldsymbol{\phi}_{t}-\boldsymbol{\phi}_{t-1}\right)\right]$$

$$\leq\frac{\left(\alpha_{t-1}-\alpha_{t}\right)\beta_{1}G}{1-\beta_{1}}\mathbb{E}\left[\|\boldsymbol{m}_{t-1}\|\right]-\alpha_{t}\mathbb{E}\left[\nabla f\left(\boldsymbol{\theta}_{t-1}\right)^{\top}\frac{\boldsymbol{g}_{t}}{\max\left\{\sqrt{\boldsymbol{v}_{t-1}},\epsilon\right\}}\right] \tag{112}$$

$$\leq\frac{\left(\alpha_{t-1}-\alpha_{t}\right)\beta_{1}G}{1-\beta_{1}}\mathbb{E}\left[\|\boldsymbol{m}_{t-1}\|\right]-\alpha_{t}\sum_{i=1}^{D}\mathbb{E}\left[\frac{\left(\nabla f\left(\boldsymbol{\theta}_{t-1}\right)\right)_{i}\cdot\left(\boldsymbol{g}_{t}\right)_{i}}{\max\left\{\sqrt{\left(\boldsymbol{v}_{t-1}\right)_{i}},\epsilon\right\}}\right] \tag{113}$$

$$\leq\frac{\left(\alpha_{t-1}-\alpha_{t}\right)\beta_{1}G}{1-\beta_{1}}\mathbb{E}\left[\|\boldsymbol{m}_{t-1}\|\right]-\alpha_{t}\sum_{i=1}^{D}\mathbb{E}\left[\frac{\left(\nabla f\left(\boldsymbol{\theta}_{t-1}\right)\right)_{i}^{2}}{\max\left\{\sqrt{\left(\boldsymbol{v}_{t-1}\right)_{i}},\epsilon\right\}}\right] \tag{114}$$

$$\leq\frac{\left(\alpha_{t-1}-\alpha_{t}\right)\beta_{1}G}{1-\beta_{1}}\mathbb{E}\left[\|\boldsymbol{m}_{t-1}\|\right]-\alpha_{t}\sum_{i=1}^{D}\mathbb{E}\left[\frac{\left(\nabla f\left(\boldsymbol{\theta}_{t-1}\right)\right)_{i}^{2}}{\sqrt{\left(\boldsymbol{v}_{t-1}\right)_{i}+\epsilon^{2}}}\right] \tag{115}$$

$$\leq\frac{\left(\alpha_{t-1}-\alpha_{t}\right)\beta_{1}G}{1-\beta_{1}}\mathbb{E}\left[\|\boldsymbol{m}_{t-1}\|\right]-\alpha_{t}\mathbb{E}\left[\frac{\|\nabla f\left(\boldsymbol{\theta}_{t-1}\right)\|^{2}}{\sqrt{\sum_{i=1}^{D}\left(\boldsymbol{v}_{t-1}\right)_{i}+\epsilon^{2}}}\right] \tag{116}$$

$$\leq\frac{\left(\alpha_{t-1}-\alpha_{t}\right)\beta_{1}G}{1-\beta_{1}}\mathbb{E}\left[\|\boldsymbol{m}_{t-1}\|\right]-\alpha_{t}\frac{\mathbb{E}\left[\|\nabla f\left(\boldsymbol{\theta}_{t-1}\right)\|_{i}^{4/3}\right]^{3/2}}{\sqrt{\mathbb{E}\left[\sum_{i=1}^{D}\left(\boldsymbol{v}_{t-1}\right)_{i}\right]+\epsilon^{2}}} \tag{117}$$

$$\leq\frac{\left(\alpha_{t-1}-\alpha_{t}\right)\beta_{1}G}{1-\beta_{1}}\mathbb{E}\left[\|\boldsymbol{m}_{t-1}\|\right]-\alpha_{t}\frac{\mathbb{E}\left[\|\nabla f\left(\boldsymbol{\theta}_{t-1}\right)\|_{i}^{4/3}\right]^{3/2}}{\sqrt{\left(1-\beta_{2}^{t}\right)G^{2}+\epsilon^{2}}} \tag{118}$$

$$\leq\frac{\left(\alpha_{t-1}-\alpha_{t}\right)\beta_{1}\left(1-\beta_{1}^{t-1}\right)G^{2}}{\left(1-\beta_{1}\right)\sqrt{\beta_{2}^{t-2}+\epsilon^{2}}}-\alpha_{t}\frac{\mathbb{E}\left[\|\nabla f\left(\boldsymbol{\theta}_{t-1}\right)\|_{i}^{4/3}\right]^{3/2}}{\sqrt{\left(1-\beta_{2}^{t}\right)G^{2}+\epsilon^{2}}}. \tag{119}$$

$\square$

**Lemma G.10.** *For the ADOPT algorithm, the following holds for $t\geq0$:*

$$\mathbb{E}\left[\sum_{i=1}^{D}\left(\boldsymbol{v}_{t}\right)_{i}\right]\leq\left(1-\beta_{2}^{t}\right)G^{2}. \tag{120}$$

*Proof.*

$$\mathbb{E}\left[\sum_{i=1}^{D}\left(\boldsymbol{v}_{t}\right)_{i}\right]=\mathbb{E}\left[\left(1-\beta_{2}\right)\sum_{i=1}^{D}\sum_{k=1}^{t}\beta_{2}^{t-k}\left(\boldsymbol{g}_{k-1}\right)_{i}^{2}\right] \tag{121}$$

$$\leq\left(1-\beta_{2}\right)G^{2}\sum_{k=1}^{t}\beta_{2}^{t-k} \tag{122}$$

$$=\left(1-\beta_{2}^{t}\right)G^{2}. \tag{123}$$

$\square$

**Lemma G.11.** *For the ADOPT algorithm, the following holds for $0\leq t\leq T$.*

$$\mathbb{E}\left[\|\boldsymbol{m}_{t}\|^{2}\right]\leq\frac{G^{2}}{\epsilon^{2}}. \tag{124}$$

*Proof.*

$$\mathbb{E}\left[\|\boldsymbol{m}_t\|^2\right]$$

$$= \mathbb{E}\left[\left\|\beta_1 \boldsymbol{m}_{t-1} + (1-\beta_1)\frac{\boldsymbol{g}_t}{\max\left\{\sqrt{\boldsymbol{v}_{t-1}},\epsilon\right\}}\right\|^2\right] \tag{125}$$

$$= \mathbb{E}\left[\beta_1^2 \|\boldsymbol{m}_{t-1}\|^2 + (1-\beta_1)^2\left\|\frac{\boldsymbol{g}_t}{\max\left\{\sqrt{\boldsymbol{v}_{t-1}},\epsilon\right\}}\right\|^2 + 2\beta_1(1-\beta_1)\boldsymbol{m}_{t-1}^\top\frac{\boldsymbol{g}_t}{\max\left\{\sqrt{\boldsymbol{v}_{t-1}},\epsilon\right\}}\right] \tag{126}$$

$$\leq \mathbb{E}\left[\beta_1 \|\boldsymbol{m}_{t-1}\|^2 + (1-\beta_1)\left\|\frac{\boldsymbol{g}_t}{\max\left\{\sqrt{\boldsymbol{v}_{t-1}},\epsilon\right\}}\right\|^2\right] \tag{127}$$

$$\leq \mathbb{E}\left[\beta_1 \|\boldsymbol{m}_{t-1}\|^2 + \frac{1-\beta_1}{\epsilon^2}\|\boldsymbol{g}_t\|^2\right] \tag{128}$$

$$\leq \mathbb{E}\left[\frac{1-\beta_1}{\epsilon^2}\sum_{k=1}^{t}\beta_1^{t-k}\|\boldsymbol{g}_k\|^2\right] \tag{129}$$

$$\leq \frac{(1-\beta_1)G^2}{\epsilon^2}\sum_{k=1}^{t}\beta_1^{t-k} \tag{130}$$

$$\leq \frac{(1-\beta_1^t)G^2}{\epsilon^2} \tag{131}$$

$$\leq \frac{G^2}{\epsilon^2}. \tag{132}$$

First inequality is derived using the following fact:

$$\forall \lambda > 0, \boldsymbol{x}, \boldsymbol{y} \in \mathbb{R}^d, \boldsymbol{x}^\top \boldsymbol{y} \leq \frac{\lambda}{2}\|\boldsymbol{x}\|^2 + \frac{1}{2\lambda}\|\boldsymbol{y}\|^2 \tag{133}$$

By setting $\lambda = (1-\beta_1)/\beta_1$, $\boldsymbol{x} = \beta_1 \boldsymbol{m}_{t-1}$, $\boldsymbol{y} = (1-\beta_1)\boldsymbol{g}_t/\max\left\{\sqrt{\boldsymbol{v}_{t-1}},\epsilon\right\}$, we obtain

$$2\beta_1(1-\beta_1)\boldsymbol{m}_{t-1}^\top\frac{\boldsymbol{g}_t}{\max\left\{\sqrt{\boldsymbol{v}_{t-1}},\epsilon\right\}} \leq \beta_1(1-\beta_1)\left(\|\boldsymbol{m}_{t-1}\|^2 + \left\|\frac{\boldsymbol{g}_t}{\max\left\{\sqrt{\boldsymbol{v}_{t-1}},\epsilon\right\}}\right\|^2\right) \tag{134}$$

Injecting it into Eq. (126), we obtain Eq. (127).

$\square$

**Lemma G.12.** *For the ADOPT algorithm, the following holds for $t \geq 0$.*

$$\mathbb{E}\left[\|\boldsymbol{m}_t\|\right] \leq \frac{G}{\epsilon} \tag{135}$$

*Proof.*

$$\mathbb{E}\left[\|\boldsymbol{m}_t\|\right] = \mathbb{E}\left[\left\|(1-\beta_1)\sum_{k=1}^{t}\beta_1^{t-k}\frac{\boldsymbol{g}_k}{\max\left\{\sqrt{\boldsymbol{v}_{k-1}},\epsilon\right\}}\right\|\right] \tag{136}$$

$$\leq (1-\beta_1)\sum_{k=1}^{t}\beta_1^{t-k}\mathbb{E}\left[\left\|\frac{\boldsymbol{g}_k}{\max\left\{\sqrt{\boldsymbol{v}_{k-1}},\epsilon\right\}}\right\|\right] \tag{137}$$

$$\leq (1-\beta_1)\sum_{k=1}^{t}\frac{\beta_1^{t-k}}{\epsilon}\mathbb{E}\left[\|\boldsymbol{g}_k\|\right] \tag{138}$$

$$\leq \frac{1-\beta_1}{\epsilon}\sum_{k=1}^{t}\beta_1^{t-k}\mathbb{E}\left[\|\boldsymbol{g}_k\|^2\right]^{1/2} \tag{139}$$

$$\leq \frac{(1-\beta_1)\,G}{\epsilon}\sum_{k=1}^{t}\beta_1^{t-k} \tag{140}$$

$$= \frac{\left(1-\beta_1^t\right)G}{\epsilon} \tag{141}$$

$$\leq \frac{G}{\epsilon}. \tag{142}$$

$\square$

**Lemma G.13.** *For the ADOPT algorithm, the following holds for $t \geq 1$:*

$$\mathbb{E}\left[\|\boldsymbol{\phi}_{t-1}-\boldsymbol{\theta}_{t-1}\|\,\|\boldsymbol{\phi}_t-\boldsymbol{\phi}_{t-1}\|\right]$$

$$\leq \frac{\alpha_{t-1}\left(\alpha_{t-1}-\alpha_t\right)\beta_1^2\left(1-\beta_1^{t-1}\right)G^2}{\epsilon^2\left(1-\beta_1\right)^2} + \frac{\alpha_t\alpha_{t-1}\beta_1\sqrt{1-\beta_1^{t-1}}G^2}{\epsilon^2\left(1-\beta_1\right)}. \tag{143}$$

*Proof.*

$$\|\boldsymbol{\phi}_{t-1}-\boldsymbol{\theta}_{t-1}\|\,\|\boldsymbol{\phi}_t-\boldsymbol{\phi}_{t-1}\|$$

$$= \left\|-\frac{\alpha_{t-1}\beta_1}{1-\beta_1}\boldsymbol{m}_{t-1}\right\|\left\|\frac{\left(\alpha_{t-1}-\alpha_t\right)\beta_1}{1-\beta_1}\boldsymbol{m}_{t-1}-\alpha_t\frac{\boldsymbol{g}_t}{\max\left\{\sqrt{\boldsymbol{v}_{t-1}},\epsilon\right\}}\right\| \tag{144}$$

$$\leq \frac{\alpha_{t-1}\beta_1}{1-\beta_1}\|\boldsymbol{m}_{t-1}\|\left(\frac{\left(\alpha_{t-1}-\alpha_t\right)\beta_1}{1-\beta_1}\|\boldsymbol{m}_{t-1}\|+\alpha_t\left\|\frac{\boldsymbol{g}_t}{\max\left\{\sqrt{\boldsymbol{v}_{t-1}},\epsilon\right\}}\right\|\right) \tag{145}$$

$$\leq \frac{\alpha_{t-1}\left(\alpha_{t-1}-\alpha_t\right)\beta_1^2}{\left(1-\beta_1\right)^2}\|\boldsymbol{m}_{t-1}\|^2 + \frac{\alpha_t\alpha_{t-1}\beta_1}{1-\beta_1}\|\boldsymbol{m}_{t-1}\|\left\|\frac{\boldsymbol{g}_t}{\max\left\{\sqrt{\boldsymbol{v}_{t-1}},\epsilon\right\}}\right\|. \tag{146}$$

Taking the expectation yields:

$$\mathbb{E}\left[\|\boldsymbol{\phi}_{t-1}-\boldsymbol{\theta}_{t-1}\|\,\|\boldsymbol{\phi}_t-\boldsymbol{\phi}_{t-1}\|\right]$$

$$\leq \frac{\alpha_{t-1}\left(\alpha_{t-1}-\alpha_t\right)\beta_1^2}{\left(1-\beta_1\right)^2}\mathbb{E}\left[\|\boldsymbol{m}_{t-1}\|^2\right] + \frac{\alpha_t\alpha_{t-1}\beta_1}{1-\beta_1}\mathbb{E}\left[\|\boldsymbol{m}_{t-1}\|\left\|\frac{\boldsymbol{g}_t}{\max\left\{\sqrt{\boldsymbol{v}_{t-1}},\epsilon\right\}}\right\|\right] \tag{147}$$

$$\leq \frac{\alpha_{t-1}\left(\alpha_{t-1}-\alpha_t\right)\beta_1^2}{\left(1-\beta_1\right)^2}\mathbb{E}\left[\|\boldsymbol{m}_{t-1}\|^2\right] + \frac{\alpha_t\alpha_{t-1}\beta_1}{\left(1-\beta_1\right)\epsilon}\mathbb{E}\left[\|\boldsymbol{m}_{t-1}\|\,\|\boldsymbol{g}_t\|\right] \tag{148}$$

$$\leq \frac{\alpha_{t-1}\left(\alpha_{t-1}-\alpha_t\right)\beta_1^2\left(1-\beta_1^{t-1}\right)G^2}{\epsilon^2\left(1-\beta_1\right)^2} + \frac{\alpha_t\alpha_{t-1}\beta_1\sqrt{1-\beta_1^{t-1}}G^2}{\left(1-\beta_1\right)\epsilon^2}. \tag{149}$$

$\square$

**Lemma G.14.** *For the ADOPT algorithm, the following holds for* $t \geq 1$:

$$
\mathbb{E}\left[\|\boldsymbol{\phi}_t - \boldsymbol{\phi}_{t-1}\|^2\right]
$$

$$
\leq \frac{(\alpha_{t-1} - \alpha_t)^2 \beta_1^2 \left(1 - \beta_1^{t-1}\right) G^2}{(1 - \beta_1)^2 \epsilon^2} + \frac{\alpha_t^2 G^2}{\epsilon^2} + \frac{\alpha_t \left(\alpha_{t-1} - \alpha_t\right) \beta_1 \sqrt{1 - \beta_1^{t-1}} G^2}{(1 - \beta_1) \epsilon^2}. \tag{150}
$$

*Proof.*

$$
\|\boldsymbol{\phi}_t - \boldsymbol{\phi}_{t-1}\|^2
$$

$$
= \left\| \frac{(\alpha_{t-1} - \alpha_t)\beta_1}{1 - \beta_1} \boldsymbol{m}_{t-1} - \alpha_t \frac{\boldsymbol{g}_t}{\max\left\{\sqrt{\boldsymbol{v}_{t-1}}, \epsilon\right\}} \right\|^2 \tag{151}
$$

$$
= \frac{(\alpha_{t-1} - \alpha_t)^2 \beta_1^2}{(1 - \beta_1)^2} \|\boldsymbol{m}_{t-1}\|^2 + \alpha_t^2 \left\| \frac{\boldsymbol{g}_t}{\max\left\{\sqrt{\boldsymbol{v}_{t-1}}, \epsilon\right\}} \right\|^2
$$

$$
- \frac{\alpha_t (\alpha_{t-1} - \alpha_t)\beta_1}{1 - \beta_1} \boldsymbol{m}_{t-1}^\top \frac{\boldsymbol{g}_t}{\max\left\{\sqrt{\boldsymbol{v}_{t-1}}, \epsilon\right\}} \tag{152}
$$

$$
\leq \frac{(\alpha_{t-1} - \alpha_t)^2 \beta_1^2}{(1 - \beta_1)^2} \|\boldsymbol{m}_{t-1}\|^2 + \frac{\alpha_t^2}{\epsilon^2} \|\boldsymbol{g}_t\|^2
$$

$$
+ \frac{\alpha_t (\alpha_{t-1} - \alpha_t)\beta_1}{1 - \beta_1} \|\boldsymbol{m}_{t-1}\| \left\| \frac{\boldsymbol{g}_t}{\max\left\{\sqrt{\boldsymbol{v}_{t-1}}, \epsilon\right\}} \right\| \tag{153}
$$

$$
\leq \frac{(\alpha_{t-1} - \alpha_t)^2 \beta_1^2}{(1 - \beta_1)^2} \|\boldsymbol{m}_{t-1}\|^2 + \frac{\alpha_t^2}{\epsilon^2} \|\boldsymbol{g}_t\|^2 + \frac{\alpha_t (\alpha_{t-1} - \alpha_t)\beta_1}{(1 - \beta_1)\epsilon} \|\boldsymbol{m}_{t-1}\| \|\boldsymbol{g}_t\|. \tag{154}
$$

Taking the expectation yields:

$$
\mathbb{E}\left[\|\boldsymbol{\phi}_t - \boldsymbol{\phi}_{t-1}\|^2\right]
$$

$$
\leq \frac{(\alpha_{t-1} - \alpha_t)^2 \beta_1^2}{(1 - \beta_1)^2} \mathbb{E}\left[\|\boldsymbol{m}_{t-1}\|^2\right] + \frac{\alpha_t^2}{\epsilon^2} \mathbb{E}\left[\|\boldsymbol{g}_t\|^2\right] + \frac{\alpha_t (\alpha_{t-1} - \alpha_t)\beta_1}{(1 - \beta_1)\epsilon} \mathbb{E}\left[\|\boldsymbol{m}_{t-1}\| \|\boldsymbol{g}_t\|\right] \tag{155}
$$

$$
\leq \frac{(\alpha_{t-1} - \alpha_t)^2 \beta_1^2 \left(1 - \beta_1^{t-1}\right) G^2}{(1 - \beta_1)^2 \epsilon^2} + \frac{\alpha_t^2 G^2}{\epsilon^2} + \frac{\alpha_t (\alpha_{t-1} - \alpha_t)\beta_1}{(1 - \beta_1)\epsilon} \mathbb{E}\left[\|\boldsymbol{m}_{t-1}\| \|\boldsymbol{g}_t\|\right] \tag{156}
$$

$$
\leq \frac{(\alpha_{t-1} - \alpha_t)^2 \beta_1^2 \left(1 - \beta_1^{t-1}\right) G^2}{(1 - \beta_1)^2 \epsilon^2} + \frac{\alpha_t^2 G^2}{\epsilon^2} + \frac{\alpha_t (\alpha_{t-1} - \alpha_t)\beta_1 \sqrt{1 - \beta_1^{t-1}} G^2}{(1 - \beta_1)\epsilon^2}. \tag{157}
$$

$$\square$$

# H   Additional Experiments

**Deep reinforcement learning:**

We train reinforcement learning (RL) agents using the soft actor crtitic algorithm [Haarnoja et al., 2018] with ADOPT for the optimizer. As a benchmark, we use a continuous control tasks of HalfCheetah-v4 on MuJoCo simulator [Todorov et al., 2012]. For comparison to ADOPT, Adam is used as a baseline optimizer. We follow the hyperparameter settings recommended by Stable-Baselines3 [Raffin et al., 2021], and just change the choice of an optimizer. The result is shown in Figure 6. The error bars indicate 95% confidence intervals of three trials. We observe slight performance improvement by using ADOPT instead of Adam.

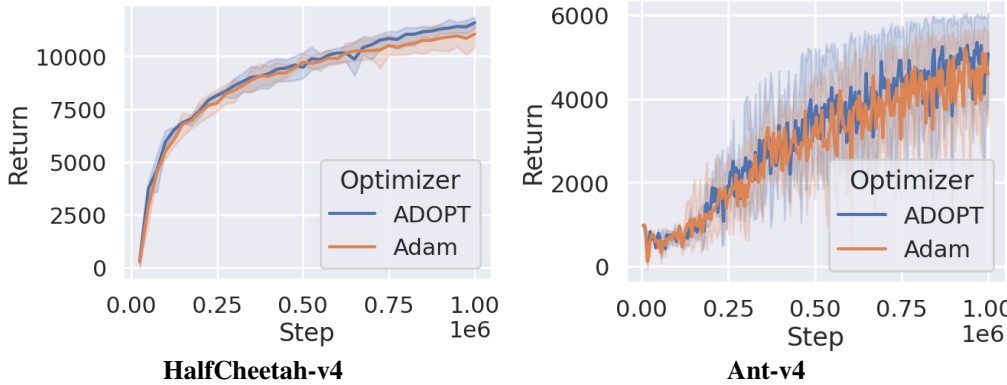

**HalfCheetah-v4**    **Ant-v4**

Figure 6: Performance comparison between Adam and ADOPT in reinforcement learning.

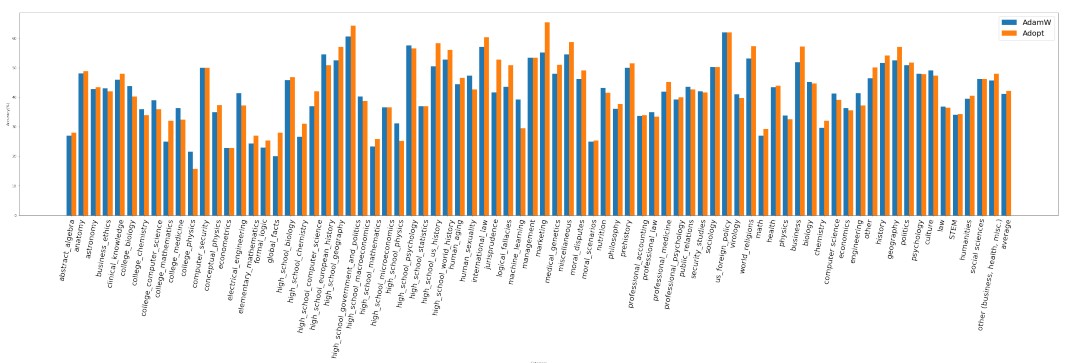

Figure 7: Comparison of MMLU scores for LLaMA-7B finetuned via instruction following using AdamW and ADOPT.

# I Details of Experimental Setups

## I.1 Code

Our implementation for the experiment is available at `https://github.com/iShohei220/adopt`.

## I.2 Total amount of compute

We run our experiments mainly on cloud GPU instances with $8\times$ A100. It took approximately 320 hours for our experiments in total.

## I.3 License of Assets

**Datasets:** The MNIST database is downloaded from `http://yann.lecun.com/exdb/mnist`, which is license-free. The terms of access for the ImageNet database is provided at `https://www.image-net.org/download`. The dataset of Stanford Alpaca is CC BY NC 4.0 (allowing only non-commercial use).

**Pretrained models:** The pretrained model of LLaMA is provided under GNU General Public License v3.0.

**Simulator:** MuJoCo is provided under Apache License 2.0.

**Code:** Our implementation of ImageNet classification is based on the Torchvision's official training recipe provided at `https://github.com/UiPath/torchvision/tree/master/references/classification`. Torchvision is provided under BSD 3-Clause License. We use the official imple-

mentation of NVAE provided at `https://github.com/NVlabs/NVAE`, whose license is described at `https://github.com/NVlabs/NVAE/blob/master/LICENSE`.

