# OpenReview forum: "ADOPT: Modified Adam Can Converge with Any $\beta_2$ with the Optimal Rate"
_NeurIPS.cc/2024/Conference — NeurIPS 2024 poster_

### Official Review · Reviewer_bGLP · 2024-07-10

**Soundness:** 3
**Presentation:** 3
**Contribution:** 3
**Rating:** 5
**Confidence:** 3

**Summary:**

The submitted work analyzes the divergence of Adam and RMSprop in smooth nonconvex settings. The authors propose a new optimizer ADOPT, whose convergence does not depend on the second-moment coefficient $\beta_2$.
The proposed optimizer is evaluated in toy settings, image classification, language modeling (finetuning settings), and reinforcement learning.

**Strengths:**

* This paper tackles the non-convergence problem of the Adam optimizer. As Adam is the go-to optimizer for deep learning, deeply understanding Adam optimizer in detail is valuable.
* The paper resolves the non-convergence issue of Adam without requiring bounded noise assumptions or specific $\beta_2$.
* The experiments are performed in diverse settings, including image classification, generative modeling, and deep reinforcement learning.

**Weaknesses:**

* The non-convergence of Adam is not an issue in practice and I am unsure about the practical implications of ADOPT over Adam. The non-convergence issue arises near the minima and most modern deep learning models are not trained to convergence.
* The paper does not thoroughly investigate the sensitivity of ADOPT optimizer parameters in practical scenarios. A more comprehensive study on the optimizer parameters and comparison with Adam is required to understand its robustness and practical utility.

**Questions:**

* In realistic settings, where is the marginal convergence speed in ADOPT coming from? My understanding is that non-convergence of Adam would matter near the minima only. Faster training way before reaching the minima is not expected. In particular, in language modeling pretraining, convergence is usually not achieved.
* Does the difference in the recommendation of the epsilon parameter for ADOPT simply arise because it is added inside the square root instead of outside?

**Limitations:**

See Weaknesses above.

---

> ### Author Rebuttal · Authors · 2024-08-07
>
> Thank you for your insightful comments. We will answer your questions to address your concerns. Please also refer to the general response and the PDF attached with it.
>
>
> > The non-convergence of Adam is not an issue in practice and I am unsure about the practical implications of ADOPT over Adam. The non-convergence issue arises near the minima and most modern deep learning models are not trained to convergence.
>
> > In realistic settings, where is the marginal convergence speed in ADOPT coming from? My understanding is that non-convergence of Adam would matter near the minima only. Faster training way before reaching the minima is not expected. In particular, in language modeling pretraining, convergence is usually not achieved.
>
> > The paper does not thoroughly investigate the sensitivity of ADOPT optimizer parameters in practical scenarios. A more comprehensive study on the optimizer parameters and comparison with Adam is required to understand its robustness and practical utility.
>
> I agree with you that in modern deep learning models, such as large language models, training is rarely run until full convergence is achieved. However, even in such cases, Adam's non-convergence can be a problem. For example, when training with a large dataset or small batch size, the constant term in Theorem 3.1 can become non-negligible due to the large gradient noise. In such cases, Adam may become unstable even in the early stages of training. To confirm that this happens empirically, we have added a new experiment of pre-training of GPT-2. See the figure in the PDF attached to the general response for the results. We observed that ADOPT is always stable in this experiment, whereas Adam actually diverges in the early phase of training due to loss spikes when the batch size is small. We also observed that ADOPT performed slightly better even when the batch size was large. We believe that these results clearly demonstrate the practical effectiveness of ADOPT in modern deep learning.
>
> > Does the difference in the recommendation of the epsilon parameter for ADOPT simply arise because it is added inside the square root instead of outside?
>
> This is a good question. In fact, the reason that ADOPT requires larger $\epsilon$ than Adam is not simply due to the order of square root and addition. This can be explained by theory: as can be seen from Theorem 3.1, the convergence bounds for Adam are of the order of $\log(\epsilon^{-2})$, while the convergence bounds for ADOPT in Eq. (33) are of the order of $\epsilon^{-2}$. In other words, when$\epsilon$ is small, the convergence bound of ADOPT becomes loose more rapidly than those of Adam. For this reason, it is safer for ADOPT to use slightly larger $\epsilon$ than Adam. However, our default settings have been found to work robustly in all experiments, so tuning of the eps is rarely needed, even in practice.
>
> We would be glad to respond to any further questions and comments that you may have.
>
> Thanks.
>
> ### References
>
> [1] github.com/karpathy/nanoGPT

---

> ### Comment · Reviewer_bGLP · 2024-08-08
>
> I thank the reviewers for their responses.
>
> > ...when training with a large dataset or small batch size, the constant term in Theorem 3.1 can become non-negligible due to the large gradient noise. In such cases, Adam may become unstable even in the early stages of training. To confirm that this happens empirically, we have added a new experiment of pre-training of GPT-2. See the figure in the PDF attached to the general response for the results. We observed that ADOPT is always stable in this experiment, whereas Adam actually diverges in the early phase of training due to loss spikes when the batch size is small....
>
> I am unsure if the statements from Theorem 3.1 generalize to realistic scenarios:
> 1.  First, the Theorem corresponds to RMSprop and there are various assumptions 2.1-2.3 and 2.5, which can become invalid in practical scenarios.
> 2. Regarding the GPT pretaining result, as the optimal learning rate range for the two optimizers can be very different, it is unclear if the learning rate for Adam was just large whereas it wasn't the case for ADOPT. A compelling result would include performance vs. learning rate for the different batch sizes. I do not expect the authors to do these experiments in this short rebuttal. If the authors have already scanned the learning rates, I would request them to point me to the result.
>
> To explicitly show that ADOPT is more stable during early training, the authors can track the top eigenvalue of the pre-conditioned Hessian [1], which determines the stability of adaptive optimizers for large batches. An alternative experiment would involve plotting heatmaps of performance against learning rate and batch size. In small-scale experiments, if the authors can show that ADOPT either (i) has a smaller pre-conditioned sharpness during early training or (ii) the optimal performance for optimal covers a large range of hyperparameters such as learning rate and batch size. Then, it would be convincing that ADOPT works reasonably well compared to Adam.
>
> [1] Adaptive Gradient Methods at the Edge of Stability, https://arxiv.org/abs/2207.14484
>
> I would like to reiterate that I do not expect the authors to perform these experiments in this discussion period.
>
> My concerns regarding the practical implications of the non-convergence issues remain. Therefore, I am keeping my current score.

---

> > ### Author Response · Authors · 2024-08-10
> > **Response to the Comment to Reviewer bGLP**
> >
> > Thank you for your comment.
> >
> > As you pointed out, Adam's loss spikes in the GPT-2 experiment may simply be due to the learning rate being too high, so we also ran the experiment with the learning rate lowered from the default of 6e-4 to 1e-4 for the case of a small batch size. The results are shown in the table below; for Adam, lowering the learning rate only slows down the timing of loss spikes and eventually causes divergence in training. ADOPT, on the other hand, is able to perform stable learning at all settings.
> > Further reductions in the learning rate resulted in significantly slower training, so we did not include them in the table.
> >
> > | Optimizer   |  LR  | 50K iters   | 100K iters     | 150K iters | 200K iters |
> > | ------------- | ---- | :----------: | :------------:  | :-----------: | :----------: |
> > | Adam         | 6e-4 | 7.64          | 7.54             |  -                |                 |
> > | Adam         | 1e-4 | 3.26          | 3.17             | 7.09           | 7.56         |
> > | ADOPT       | 6e-4 | 3.22          | 3.17             | 3.13           | 3.10         |
> > | ADOPT       | 1e-4 | 3.16          | 3.09             | 3.04           | 3.02         |
> >
> > This result is also consistent with theory: the constant term for Adam's convergence bounds in Theorem 3.1 is independent of the learning rate, so a failure in training cannot be prevented by reducing the learning rate. As you stated, Theorem 3.1 is for RMSprop, which does not account for momentum, but there is a similar constant term in Adam's convergence bounds in prior studies (e.g., [1]) that is independent of the learning rate. Thus, this result is a practical example of how Adam can become unstable when training on a large-scale data set or with a small batch size, as the gradient noise increases in such cases. Such instability is not unique to this experimental setting, as it has often been reported to be observed in the training of larger LLMs [2, 3].
> >
> > We hope that this will address your concerns about our submission.
> > We would be glad to respond to any further comments that you may have during the discussion period.
> >
> > Thanks!
> >
> >
> > ### References
> >
> > [1] Alexandre Défossez, Leon Bottou, Francis Bach, and Nicolas Usunier. A simple convergence proof of adam and adagrad. Transactions on Machine Learning Research, 2022. ISSN 2835-8856.
> >
> > [2] Molybog, Igor, et al. "A theory on adam instability in large-scale machine learning." arXiv preprint arXiv:2304.09871 (2023).
> >
> > [3] Takase, Sho, et al. "Spike No More: Stabilizing the Pre-training of Large Language Models." arXiv preprint arXiv:2312.16903 (2023).

---

> > > ### Comment · Reviewer_bGLP · 2024-08-11
> > >
> > > Thanks to the authors for the additional results. I suggested looking at the test loss vs. learning rate curves because Adam can achieve the same performance as ADOPT but with a smaller learning rate. This would just imply that ADOPT achieves optimal performance at a high learning rate. I am still unclear about the following:
> > >
> > > 1. For the optimal hyperparameters, such as learning rate, does ADOPT always perform better than Adam?
> > > 2. Does ADOPT require a smaller warmup (or no warmup)? Is warmup used in the GPT experiments?
> > > 3. Is the range of optimal learning rates longer for ADOPT compared to Adam? Or the size of the range is the same but it is shifted?

---

> ### Author Response · Authors · 2024-08-14
> **Response to the Additional Comment to Reviewer bGLP**
>
> Thank you for your comment.
>
> > For the optimal hyperparameters, such as learning rate, does ADOPT always perform better than Adam?
>
> We have summarized the result regarding the optimal hyperparameter settings in terms of the batch size and the learning rate in the table below.
> The result shows the test loss after running 100K training iterations.
> The best results are shown in bold.
> When comparing the best results, ADOPT shows a slightly better result than Adam.
>
> **Adam**
>
> | Batch size \ LR | 6e-5  |1e-4 | 6e-4        |
> | ----------------- | ------- |------ | --------   |
> | 96                     |  3.21 | 3.17 | 7.54        |
> | 480                   |     -    | 3.31 | **3.02**   |
>
> **ADOPT**
>
> | Batch size \ LR | 6e-5      |1e-4      | 6e-4 |
> | ----------------- | --------- |--------- | ----- |
> | 96                     | 3.11       |  3.09      | 3.17 |
> | 480                   | -            | **2.98** | 3.00 |
>
> > Does ADOPT require a smaller warmup (or no warmup)? Is warmup used in the GPT experiments?
>
> In the GPT-2 experiment, following the default setting of the nanoGPT code base, the linear warmup is used for the first 2000 iterations.
> Please refer to the original code provided at github.com/karpathy/nanoGPT/blob/master/train.py for more detailed settings.
> We did not change the experimental settings except for choices of the optimizer, the batch size, and the learning rate.
>
> > Is the range of optimal learning rates longer for ADOPT compared to Adam? Or the size of the range is the same but it is shifted?
>
> As can be seen from the table above, ADOPT seems to work more robustly to the choice of the learning rate compared to Adam.
>
> If you have any further comments or questions, we would be glad to respond them.
>
> Thanks!

---

### Official Review · Reviewer_AhBk · 2024-07-11

**Soundness:** 3
**Presentation:** 2
**Contribution:** 2
**Rating:** 5
**Confidence:** 4

**Summary:**

The paper titled "ADOPT: Modified Adam Can Converge with Any β2 with the Optimal Rate" introduces a new adaptive gradient method named ADOPT. This method aims to address the non-convergence issues of the Adam optimization algorithm. Adam, despite its popularity in deep learning, does not theoretically converge unless the hyperparameter β2 is chosen in a problem-dependent manner. The paper proposes ADOPT, which achieves the optimal convergence rate with any choice of β2, without relying on the bounded noise assumption. ADOPT modifies Adam by removing the current gradient from the second moment estimate and changing the order of the momentum update and normalization. The paper also presents extensive numerical experiments showing ADOPT's superior performance across various tasks, including image classification, generative modeling, natural language processing, and deep reinforcement learning.

**Strengths:**

1. The paper provides a robust theoretical foundation for the proposed ADOPT algorithm, demonstrating its convergence with an optimal rate
2. The proposed method is practically significant as it eliminates the need for problem-dependent tuning of the β2 parameter, making it more user-friendly and broadly applicable.
3. The paper includes comprehensive experiments across various tasks, showing that ADOPT consistently outperforms Adam and its variants.
4. The paper clearly identifies the non-convergence problem of Adam and provides a well-justified solution in ADOPT.

**Weaknesses:**

1. The analysis still relies on the assumption that the second moment of the stochastic gradient is uniformly bounded, which might not always hold in practice.
2. The paper could have included comparisons with more recent optimizers beyond Adam to strengthen its empirical claims, like the recent work CO2 ("CO2: Efficient distributed training with full communication-computation overlap." ICLR (2024).).
3. The paper does not thoroughly discuss the potential computational overhead introduced by ADOPT compared to Adam and other optimizers.

**Questions:**

1. How can the assumption on the second moment being uniformly bounded be relaxed? Are there any potential methods or future work suggested to address this?
2. What is the computational cost of ADOPT compared to Adam and AMSGrad? Is there a significant overhead that might affect its practicality in large-scale applications?
3. While ADOPT performs well on a variety of tasks, how does it perform on tasks not covered in the experiments? Are there specific types of problems where ADOPT might not be as effective?

**Limitations:**

see the weaknesses.

---

> ### Author Rebuttal · Authors · 2024-08-07
>
> Thank you for your insightful comments. We will answer your questions to address your concerns. Please also refer to the general response and the attached PDF.
>
>
> > The analysis still relies on the assumption that the second moment of the stochastic gradient is uniformly bounded, which might not always hold in practice.
> > How can the assumption on the second moment being uniformly bounded be relaxed? Are there any potential methods or future work suggested to address this?
>
> As you mentioned, our analysis still relies on the bounded second moment assumption, which is sometimes violated in practice, although it is milder than the bounded stochastic gradient assumption used in the previous works. A promising direction to further relax this assumption is to use the bounded variance assumption, where $\mathbb{E} [ \|| g - \nabla f \||^2 ]$ is assumed to be bounded. We mentioned it in the third paragraph of Section 6 as a limitation and future work.
>
> > The paper could have included comparisons with more recent optimizers beyond Adam to strengthen its empirical claims, like the recent work CO2 ("CO2: Efficient distributed training with full communication-computation overlap." ICLR (2024).).
>
> Thank you for your suggestion. We think that CO2 is specifically designed for efficient distributed training, so their contributions are orthogonal to ours. In our experiment, we compare other optimizers which share the motivation of addressing the non-convergence issue of Adam (e.g., AMSGrad). Of course, combining the techniques of other works (e.g., CO2) with our ADOPT could be a promising direction to further improve the performance, but we leave it for future work.
>
> > The paper does not thoroughly discuss the potential computational overhead introduced by ADOPT compared to Adam and other optimizers.
> > What is the computational cost of ADOPT compared to Adam and AMSGrad? Is there a significant overhead that might affect its practicality in large-scale applications?
>
> Thank you for your comment. Let us clarify the computational cost of ADOPT compared to Adam and AMSGrad. The computational cost of ADOPT is equal to that of Adam and less than that of AMSGrad. Since both Adam and ADOPT need to store momentum $m_t$ and second-order moments $v_t$, their memory costs are about the same. In addition, Adam computes bias corrections when updating parameters, while our implementation of ADOPT omits them, making ADOPT slightly less computationally expensive than Adam. On the other hand, AMSGrad requires storing $\hat{v}_t$ in addition to $m_t$ and $v_t$, so its memory cost is larger than ADOPT (and Adam).
>
> > While ADOPT performs well on a variety of tasks, how does it perform on tasks not covered in the experiments? Are there specific types of problems where ADOPT might not be as effective?
>
> As far as we have experimented, ADOPT has always shown better results than Adam. To further strengthen our experimental results, we also performed pre-training experiments on GPT-2 and found that ADOPT performed better than Adam in those experiments as well. In particular, we observed that Adam suffers from loss spikes when the batch size is small and the gradient noise is large, while ADOPT always performs well stably. See the general response for more details.
>
> We would be glad to respond to any further questions and comments that you may have.
>
> Thanks.

---

> ### Author Response · Authors · 2024-08-14
> **A Gentle Reminder to Reviewer AhBk**
>
> Thank you again for your efforts in reviewing our paper and your constructive comments. The discussion period will end soon, so please let us know if you have further comments about our reply to your feedback.
>
> Thanks.

---

### Official Review · Reviewer_qABQ · 2024-07-12

**Soundness:** 3
**Presentation:** 3
**Contribution:** 2
**Rating:** 6
**Confidence:** 4

**Summary:**

Motivated by the counterexample due to Reddi et al., this work designs an adaptive optimizer that can converge for the choice of beta_2 that is independent of the problem instance. Their analysis works for a more general condition than the previous works.

**Strengths:**

It has overall good presentation. The main scope and results are presented well.

**Weaknesses:**

The theoretical results look sound. I have some questions below.

**Questions:**

- In Theorem 4.1, please specify the choice of hyperparameters (beta_1, beta_2, \eps). I could not find this even in the appendix, even though the main text said it can be found there. In particular, does the choice found by your analysis similar to what people use in practice?

- If the choice of beta's do not match with your theoretical prediction, I think the claim that the new optimizer AdaOPT alleviates the parameter-tuning is not an overclaim. At the end of the day, it seems that the authors have to tune those parameters depending on the settings. Hence, practically speaking, there's no reason for practitioners to use the proposed algorithm over the original Adam.

- It seems that for your algorithm, $\epsilon$ needs to be chosen much larger than that of Adam. How sensitive is this choice? Why is this the case? $\epsilon$ really should be for numerical stability in the original Adam. Does your analysis suggest such a large value for $\epsilon$?

**Limitations:**

Overall, the main contribution in this work seems quite marginal. For a submission of this type, I would support a clear acceptance if
- (i) the theoretical guarantees have a noticeable innovation over the previous ones.
- (ii) or the proposed method works much better than the previous ones.

It seems that the theoretical improvement is the removal of the uniformly upper bounded assumption of stochastic gradients.
To me, this looks like a minor improvement over the previous work.
Moreover, the resulting algorithm does not seem to have major advantages over Adam. In particular, unlike the claim made in this paper, I don't think the resulting algorithm alleviates the hyper-parameter tuning. I don't see why practitioners should choose this algorithm over Adam.
Therefore, I recommend "borderline accept" for this work.

---

> ### Author Rebuttal · Authors · 2024-08-07
>
> Thank you for your insightful comments. We will answer your questions to address your concerns. Please also refer to the general response and the PDF attached with it.
>
>
> > In Theorem 4.1, please specify the choice of hyperparameters (beta_1, beta_2, \eps). I could not find this even in the appendix, even though the main text said it can be found there. In particular, does the choice found by your analysis similar to what people use in practice?
>
> > If the choice of beta's do not match with your theoretical prediction, I think the claim that the new optimizer AdaOPT alleviates the parameter-tuning is not an overclaim. At the end of the day, it seems that the authors have to tune those parameters depending on the settings. Hence, practically speaking, there's no reason for practitioners to use the proposed algorithm over the original Adam.
>
> The concrete convergence bound is provided in Eq. (33) in Appendix E, which shows that the convergence rate is $O ( 1 / \sqrt{T} )$ with any $(\beta_1, \beta_2, \epsilon)$. The bound gets tighter when $\beta_2$ is chosen to be close to 1, which corresponds to practical choices. In fact, as shown in Figure 1, the training with ADOPT tends to be stable when $\beta_2$ is close to 1, although the convergence is achieved even with small $\beta_2$.
>
> In terms of $\beta_1$, there is a gap between theory and practice. In theory, the bound will be tighter when $\beta_1$ is small, whereas $\beta_1 = 0.9$ is used in practice. This gap is consistently observed in the literature on the convergence analysis of Adam (e.g., [2, 3]). To the best of our knowledge, the effectiveness of momentum in Adam-type optimizers is still an open question.
>
> > It seems that for your algorithm,  needs to be chosen much larger than that of Adam. How sensitive is this choice? Why is this the case?  really should be for numerical stability in the original Adam. Does your analysis suggest such a large value for ?
>
> You are correct that ADOPT requires a larger $\epsilon$ than Adam. This can be explained by theory: as can be seen from Theorem 3.1, the convergence bounds for Adam are of the order of $\log(\epsilon^{-2})$, while the convergence bounds for ADOPT in Eq. (33) are of the order of $\epsilon^{-2}$. In other words, when $\epsilon$ gets small, the convergence bounds of ADOPT become loose more rapidly than those of Adam. For this reason, it is safer for ADOPT to use a slightly larger value for $\epsilon$ than Adam. Our default settings have been found to work robustly in all experiments, so tuning of the $\epsilon$ is rarely needed, even in practice.
>
> > Overall, the main contribution in this work seems quite marginal. For a submission of this type, I would support a clear acceptance if
>
> > (i) the theoretical guarantees have a noticeable innovation over the previous ones.
>
> > (ii) or the proposed method works much better than the previous ones.
>
> > It seems that the theoretical improvement is the removal of the uniformly upper bounded assumption of stochastic gradients. To me, this looks like a minor improvement over the previous work. Moreover, the resulting algorithm does not seem to have major advantages over Adam. In particular, unlike the claim made in this paper, I don't think the resulting algorithm alleviates the hyper-parameter tuning. I don't see why practitioners should choose this algorithm over Adam.
>
> Thank you for your suggestions. To demonstrate the practical effectiveness of our ADOPT more clearly, we have added an experiment in pre-training language models, in which Adam tends to suffer from optimization difficulties like loss spikes. In this experiment, we used the nanoGPT [1] code base to run the GPT-2 pre-training in OpenWebText. We observed that Adam suffered from loss spikes and completely failed the training when the batch size was small, while ADOPT was able to train stably even with small batch sizes. We also observed that ADOPT performed slightly better than Adam even for large batch sizes. This result is consistent with theory, since, when the batch size is small, the gradient noise is larger and the constant term of Adam's convergence bound in Theorem 3.1 also gets larger. Thus, the experimental results confirm that ADOPT has an advantage over Adam even in practical cases such as pre-training language models. We hope these results will address your concerns.
>
> We would be glad to respond to any further questions and comments that you may have.
>
> Thanks.
>
>
> ### References
>
> [1] github.com/karpathy/nanoGPT
>
> [2] Alexandre Défossez, Leon Bottou, Francis Bach, and Nicolas Usunier. A simple convergence proof of adam and adagrad. Transactions on Machine Learning Research, 2022. ISSN 2835-8856.
>
> [3] Bohan Wang, Jingwen Fu, Huishuai Zhang, Nanning Zheng, and Wei Chen. Closing the gap between the upper bound and lower bound of adam’s iteration complexity. In Thirty-seventh Conference on Neural Information Processing Systems, 2023.

---

> > ### Comment · Reviewer_qABQ · 2024-08-09
> > **Thanks**
> >
> > Thanks for your responses.
> > I read through them, and it partially addresses my concerns.
> > Hence, I'll increase the score to 6.
> >
> > On a side note, it seems that the benefits of momentum seems to be theoretically justified in the nonsmooth and nonconvex setting (as shown by [1]), and the authors might find it helpful to further clarify the main contributions of this work.
> >
> > [1] Ahn and Cutkosky, "Adam with model exponential moving average is effective for nonconvex optimization" (https://arxiv.org/pdf/2403.02648)

---

> ### Author Response · Authors · 2024-08-14
> **Response to the Additional Comment by Reviewer qABQ**
>
> Thank you for your additional comments.
>
> > On a side note, it seems that the benefits of momentum seems to be theoretically justified in the nonsmooth and nonconvex setting (as shown by [1]), and the authors might find it helpful to further clarify the main contributions of this work.
>
> We were not aware of the result of the paper you mentioned, so thank you for letting us know.
> As you pointed out, that paper does indeed partially explain the role of momentum in Adam, but there appear to be some limitations: first, their analysis is limited to the case $\beta_2=\beta_1^2$, which deviates from the practical choice. In addition, their analysis can only be applied to the case where Adam's update is clipped, which also deviates from the practical algorithm. Thus, the role of momentum in Adam is not yet fully understood and seems to be an open question.
> We promise to cite that paper and describe this point clearly in the final version.
>
> If you have any further comments or questions, we would be glad to respond them.
>
> Thanks.

---

### Official Review · Reviewer_f5HN · 2024-07-15

**Soundness:** 3
**Presentation:** 3
**Contribution:** 2
**Rating:** 6
**Confidence:** 3

**Summary:**

The paper proposes a new adaptive gradient method called ADOPT, which addresses the non-convergence issue of popular methods like Adam and RMSprop.  The method modifies the calculation of second moment estimates and the order of momentum calculation and scaling operations. Extensive numerical experiments demonstrate that ADOPT achieves competitive or superior results compared to existing methods across various tasks.

**Strengths:**

The paper introduces a new adaptive gradient method ADOPT that is as easy as the implementation of Adam, and enjoys easy convergence proofs.

The paper gives in-depth analysis for the convergence of ADOPT with toy examples, in comparison with the failure cases of Adam.

The paper conducts comprehensive numerical experiments on various tasks, demonstrating the competitive performance of ADOPT
 compared to the widely used Adam.

**Weaknesses:**

The convergence of a modified version of Adam is not significant from theoretical sense unless the ADOPT can beat the performance of Adam in practice given existing convergence proofs of Adam.

From the empirical results, the performance of ADOPT is not superior over Adam very much. People may be reluctant to use ADOPT in practice.

More importantly, the algorithm 1 (ADOPT) seems to require storage of three parts: g_t, m_t, v_t, which is more than what the standard Adam requires. This is quite significant drawback of ADOPT if this cannot be optimized.

------------------------------------------------------------------------------------------------- comments after rebuttal -------------------------------------------------------------The authors' response clearly resolved the memory cost concerns. I would like to increase the score to 6.

**Questions:**

see weakness.

**Limitations:**

Yes, The authors have adequately addressed the limitations.

---

> ### Author Rebuttal · Authors · 2024-08-07
>
> Thank you for your insightful comments. We will answer your questions to address your concerns. Please also refer to “Author Rebuttal by Authors” and the attached PDF.
>
> > The convergence of a modified version of Adam is not significant from theoretical sense unless the ADOPT can beat the performance of Adam in practice given existing convergence proofs of Adam.
>
> > From the empirical results, the performance of ADOPT is not superior over Adam very much. People may be reluctant to use ADOPT in practice.
>
> Your point is that ADOPT does not seem to perform so well compared to Adam in the practical experiments in Section 5 except for the toy examples. In fact, what our theory shows is that Adam fails catastrophically when the constant term of Adam's convergence bound in Theorem 3.1 becomes large; hence, in a correctly tuned situation in practice (e.g., with a sufficiently large $\beta_2$), it is natural that ADOPT will not improve significantly relative to Adam.
>
> However, to show that even in practice Adam can make catastrophic failures and ADOPT can avoid them, we have added an experiment in pre-training language models. In this experiment, we used the nanoGPT [1] code base to run the GPT-2 pre-training with OpenWebText. See the general response and the attached PDF for more details. We observed that Adam suffered from loss spikes and completely failed the training when the batch size was small, while ADOPT was able to train stably even with small batch sizes. We also observed that ADOPT performed slightly better than Adam even for large batch sizes. This result is consistent with theory, since the gradient noise is larger and the constant term of Adam's convergence bound is larger when the batch size is small. Thus, the experimental results confirm that ADOPT has an advantage over Adam even in practical cases such as pre-training language models. We hope these results will address your concerns.
>
> > More importantly, the algorithm 1 (ADOPT) seems to require storage of three parts: g_t, m_t, v_t, which is more than what the standard Adam requires. This is quite significant drawback of ADOPT if this cannot be optimized.
>
> I respectfully point out that this is a misunderstanding: the memory cost of ADOPT is equivalent to that of Adam. Only $m_t$ and $v_t$ need to be stored in ADOPT, and $g_t$ can be discarded once those updates are done. Presumably, this misunderstanding arises from the fact that $g_{t+1}$ is used to update $m_{t+1}$ in Algorithm 1, but this does not mean that $g_t$ needs to be stored. To clarify this, an equivalent alternative representation is given in Algorithm 2 in the attached PDF of the general response. In fact, experimental results confirm that the memory costs of ADOPT and Adam are equivalent. For example, the memory cost for pre-training GPT-2 is approximately 18 GB per GPU for both.
>
> We would be glad to respond to any further questions and comments that you may have.
>
> Thanks.
>
> ### References
>
> [1] github.com/karpathy/nanoGPT

---

> ### Author Response · Authors · 2024-08-14
> **A Gentle Reminder to Reviewer f5HN**
>
> Thank you again for your efforts in reviewing our paper and your constructive comments. The discussion period will end soon, so please let us know if you have further comments about our reply to your feedback.
>
> Thanks.

---

### Author Rebuttal · Authors · 2024-08-07

We thank all reviewers for their comments. They are insightful and help us to make our paper better. We have added new experiments and explanations to address the reviewer's concerns. Please also refer to the individual responses to each reviewer.

## Additional experiments of pre-training GPT-2
Since many reviewers seem to have concerns about the practical effectiveness of ADOPT, we have added a new experiment to reinforce it. In this experiment, we ran a pre-training of GPT-2 using the nanoGPT [1] code base to compare Adam and ADOPT. We used OpenWebText as the training data. Experimental setup conformed to the default settings of nanoGPT except for the selection of the optimizer. We also tested a case in which the total batch size was changed from 480 to 96, as a setting where the gradient noise becomes larger. The results are summarized in Figure 7 of the attached PDF file. The most notable finding is that in the small batch size case, Adam causes loss spikes in the early stages of training and fails to converge, while ADOPT is always able to train stably. This is consistent with Adam's theory of non-convergence. As the gradient noise increases, $G$ in Theorem 3.1 also increases, and the constant term in Adam's convergence bounds becomes non-negligible especially when using a large-scale dataset like OpenWebText. As a result, Adam is more likely to fail to train in such cases. Our ADOPT, on the other hand, does not suffer from this problem because it can always guarantee convergence. We also observed that both Adam and ADOPT work well when the batch size is large, but even in this case, ADOPT performs slightly better.


## Clarification of memory cost of ADOPT
Some reviewers have raised concerns about the memory cost of ADOPT, so we will address them as well.
The memory cost of ADOPT is exactly the same as that of Adam, with $m_t$ and $v_t$ as the two parameters to be stored during training; AMSGrad requires storage of $\hat{v}_t$ in addition to these, making its memory cost larger than that of ADOPT and Adam. We also experimentally confirmed that ADOPT and Adam have the same memory consumption during training in the GPT-2 experiment. We observed that both ADOPT and Adam require 18 GB per GPU during training.

We hope that it will address the reviewers' concerns. We would be glad to respond to any further questions and comments that you may have.

Thanks.

References

[1] github.com/karpathy/nanoGPT

---

### Decision · Program_Chairs · 2024-09-25

**Decision:**

Accept (poster)

**Comment:**

The paper introduces a new adaptive gradient method named ADOPT, which is designed to address the non-convergence issues of Adam, particularly when the $\beta_2$ parameter is not chosen in a problem-dependent manner. ADOPT modifies Adam in the computation of the second moment estimates and the order of momentum updates. It is claimed that ADOPT achieves optimal convergence rates without relying on the assumption that the gradient noise is uniformly bounded.

The reviewers acknowledged the theoretical contribution of ADOPT, though the contribution is marginal; for example, the convergence still relies on the assumption that the second moment of the stochastic gradient is uniformly bounded. Besides, the reviewers also agreed that ADOPT could be beneficial in specific scenarios, as shown by experiments. Overall, the reviewers are mostly supportive of this paper, and I think this paper could be accepted.